# TorsinB overexpression prevents abnormal twisting in DYT1 dystonia mouse models

Jay Li[1,2†], Chun-Chi Liang[3†], Samuel S Pappas[4*], William T Dauer[3,4*]

[1]Medical Scientist Training Program, University of Michigan, Ann Arbor, United States; [2]Cellular and Molecular Biology Graduate Program, University of Michigan, Ann Arbor, United States; [3]Department of Neurology, University of Michigan, Ann Arbor, United States; [4]Peter O'Donnell Jr. Brain Institute, Departments of Neuroscience and Neurology & Neurotherapeutics, University of Texas Southwestern, Dallas, United States

**Abstract** Genetic redundancy can be exploited to identify therapeutic targets for inherited disorders. We explored this possibility in DYT1 dystonia, a neurodevelopmental movement disorder caused by a loss-of-function (LOF) mutation in the *TOR1A* gene encoding torsinA. Prior work demonstrates that torsinA and its paralog torsinB have conserved functions at the nuclear envelope. This work established that low neuronal levels of torsinB dictate the neuronal selective phenotype of nuclear membrane budding. Here, we examined whether torsinB expression levels impact the onset or severity of abnormal movements or neuropathological features in DYT1 mouse models. We demonstrate that torsinB levels bidirectionally regulate these phenotypes. Reducing torsinB levels causes a dose-dependent worsening whereas torsinB overexpression rescues torsinA LOF-mediated abnormal movements and neurodegeneration. These findings identify torsinB as a potent modifier of torsinA LOF phenotypes and suggest that augmentation of torsinB expression may retard or prevent symptom development in DYT1 dystonia.

**\*For correspondence:**
samuel.pappas@utsouthwestern.edu (SSP);
william.dauer@utsouthwestern.edu (WTD)

[†]These authors contributed equally to this work

**Competing interests:** The authors declare that no competing interests exist.

## Introduction

DYT1 dystonia is a dominantly inherited movement disorder that is caused by 3 bp in-frame deletion (ΔE mutation) in the *TOR1A* gene that encodes the torsinA protein (*Ozelius et al., 1997*). Only ~30% of mutation carriers exhibit symptoms, which vary in severity from mild to severely debilitating (*Albanese et al., 2011*; *Akbari et al., 2012*). Treatments include deep brain stimulation, which is invasive, and anticholinergic drugs, which provide incomplete relief and are plagued by side effects (*Saunders-Pullman et al., 2002*; *Vidailhet et al., 2005*). These treatments suppress symptoms; no therapies are based on disease pathogenesis or alter the emergence of symptoms.

TorsinA is a nuclear envelope/endoplasmic reticulum (NE/ER) resident AAA+ protein (ATPase Associated with diverse cellular Activities) (*Ozelius et al., 1997*). Multiple lines of evidence demonstrate that the DYT1 mutation impairs torsinA function (*Tanabe et al., 2009*; *Liang et al., 2014*; *Weisheit and Dauer, 2015*; *Goodchild and Dauer, 2005*). The DYT1 mutation reduces protein stability and impairs interaction with cofactors (LAP1 and LULL1) that appear important for torsinA ATPase activity (*Goodchild and Dauer, 2005*; *Naismith et al., 2009*; *Zhao et al., 2013*).

Prior work demonstrates conserved functions for torsinA and torsinB. Their sequences are 68% identical and 85% similar, and they share cofactors LAP1 and LULL1 (*Ozelius et al., 1999*; *Goodchild and Dauer, 2005*; *Brown et al., 2014*; *Laudermilch et al., 2016*). TorsinA null mice and mice homozygous for the DYT1 mutation exhibit neural-selective abnormalities of NE structure (NE 'budding') (*Goodchild et al., 2005*; *Kim et al., 2010*). Several observations suggest that this neural

specificity results from markedly lower levels of torsinB in neurons compared with non-neuronal cells. The appearance of neuronal NE budding in torsinA mutants coincides with lower levels of torsinB during early brain maturation (*Tanabe et al., 2016*). shRNA knockdown of torsinB in torsinA null non-neuronal cells recapitulates the 'neuronal-like' NE budding phenotype (*Kim et al., 2010*). Moreover, conditional CNS deletion of both torsinA and torsinB causes NE budding in neuronal and non-neuronal (e.g. glia) cells, and overexpressing torsinB significantly reduces NE budding in torsinA null developing neurons in vitro (*Tanabe et al., 2016*).

Based on these data, we hypothesized that altering torsinB levels would modulate motor and neuropathological phenotypes of DYT1 mouse models. We pursued epistatic analyses of torsinA loss-of-function (LOF) and both torsinB reduction and overexpression, assessing established torsinA-related neuropathological and behavioral phenotypes. We demonstrate that torsinB bidirectionally modifies these phenotypes. Reducing levels of torsinB in DYT1 models (*Tor1a$^{-/-}$* or *Tor1a$^{\Delta E/-}$*) causes a dose-dependent worsening of behavioral and neuropathological phenotypes. Conversely, overexpressing torsinB from the ROSA26 locus dramatically reduces the emergence of motor and neuropathological phenotypes in two DYT1 models. Our findings demonstrate that torsinB is a genetic modifier of torsinA LOF disease-related phenotypes and suggest that enhancing torsinB function may be a viable therapeutic strategy in DYT1 dystonia.

## Results

### TorsinB null mice exhibit no apparent organismal or neuropathological phenotypes

To study the role of torsinB as a modifier of torsinA LOF, we first assessed whether torsinB null mice exhibit any pathological phenotypes. Intercrosses of *Tor1b$^{+/-}$* mice yield *Tor1b$^{-/-}$* mice that are indistinguishable from wild-type littermate controls. These mutants gain weight similarly to littermate controls (*Figure 1—figure supplement 1A*), and do not exhibit brain abnormalities when assessed by Nissl stain (*Figure 1—figure supplement 1B*). Glial fibrillary acidic protein (GFAP) immunohistochemistry did not demonstrate any areas of gliosis (*Figure 1—figure supplement 1C*). *Tor1b$^{-/-}$* mice exhibit normal cortical thickness (*Figure 1—figure supplement 1D*) and do not display abnormal limb clasping during tail suspension (*Figure 1—figure supplement 1E*). These findings enabled us to assess the role of torsinB levels on torsinA LOF-related phenotypes without the confound of additive abnormalities.

### TorsinB deletion worsens torsinA-related motor and neuropathological phenotypes

*Emx1-Cre* expresses in cortical and hippocampal excitatory neurons and glia (*Gorski et al., 2002*). This Cre field includes the motor cortex, which is a critical component of the forebrain cortico-striatal network strongly implicated in the expression of dystonic symptoms (*Ikoma et al., 1996*; *Gilio et al., 2003*; *Gilbertson et al., 2019*). Disruption of this network has been observed in several dystonia mouse models (*Maltese et al., 2014*; *Sciamanna et al., 2014*; *Maltese et al., 2017*; *Maltese et al., 2018*), and numerous clinical electrophysiological findings associate cortical dysfunction to dystonic movements in humans (*Ridding et al., 1995*; *Edwards et al., 2003*; *McDonnell et al., 2007*; *Bara-Jimenez et al., 1998*; *Bara-Jimenez et al., 2000*; *Aglioti et al., 2003*; *Quartarone et al., 2003*; *Edwards et al., 2006*). *Emx1-Cre* conditional deletion of torsinA (Emx1 (A)-CKO, *Table 1*, *Figure 1—figure supplement 2A*) significantly reduces cortical thickness (*Figure 1D*), but does not significantly alter the number of CUX1+ (marker for cortical layer II-IV) or CTIP2+ (marker for cortical layer V-VI) cortical neurons (*Figure 1E–F*; *Arlotta et al., 2005*; *Ferrere et al., 2006*). Emx1(A)-CKO mice appear grossly normal, and only a subset of these mice exhibit limb clasping during tail suspension (*Table 1*, *Figure 1F*; *Liang et al., 2014*). Consistent with prior studies, conditionally deleting both torsinA and torsinB with *Emx1-Cre* (Emx1(A+B)-CKO, *Figure 1—figure supplement 2A*) does not cause overt brain structural abnormalities at birth (*Goodchild et al., 2005*): cortical thickness and the number of CTIP2+ cells do not differ significantly from littermate controls at postnatal day 0 (P0) (*Figure 1—figure supplement 2B–C*). At P28, however, Emx1(A+B)-CKO mice exhibit significant cell loss (*Figure 1A*) and profound gliosis (*Figure 1B*) in Cre expressing regions (cerebral cortex and hippocampus). The hindbrain, which is not in the Cre

**Table 1.** Features of mouse models used in this study.

Characteristics of DYT1 dystonia mouse models including extent of Cre field, behavioral phenotype, and neuropathological phenotype.

| Model | Genotype | Cre field | Organismal/behavioral phenotype | Histologic findings | Rationale for use | Impact of torsinB modulation |
|---|---|---|---|---|---|---|
| Emx1 (A)-CKO | Emx1-Cre; Tor1a$^{KO/flx}$ | Forebrain excitatory, prominent in cortex and hippocampus | Normal appearing Limb clasping in a subset of mice | Forebrain-selective neurodegeneration | Forebrain motor loop involvement Mild behavioral and neuropathology findings to assess combined torsinA and B LOF | Reduction: Worsened neuropathology and behavior |
| Emx1-SKI | Emx1-Cre; Tor1a$^{\Delta E/flx}$ | Forebrain excitatory, prominent in cortex and hippocampus | Normal appearing Limb clasping in a subset of mice | Forebrain-selective neurodegeneration milder than that seen in Emx1-CKO | Forebrain motor loop involvement Presence of disease mutant torsinA | Reduction: Dose-dependent worsening of neuropathology and behavior |
| Nes (A)-CKO | Nestin-Cre; Tor1a$^{KO/flx}$ | Entire nervous system | Lack of postnatal weight gain Early lethality by 3rd postnatal week Overtly abnormal postures at rest | Degeneration in multiple sensorimotor regions | Clear and robust phenotypes Widespread involvement of nervous system | Overexpression: Prevention of lethality, restored weight gain Prevention of degeneration and gliosis |
| Nes-SKI | Nestin-Cre; Tor1a$^{\Delta E/flx}$ | Entire nervous system | Reduced postnatal weight gain Overt postural and developmental phenotypes at rest | Degeneration in multiple sensorimotor regions | Widespread involvement of nervous system Presence of disease mutant torsinA | Overexpression: Restored weight gain Prevention of degeneration and gliosis |
| Dlx(A)-CKO | Dlx5/6-Cre; Tor1a$^{KO/flx}$ | Forebrain GABAergic and cholinergic, including all striatal neurons | Limb clasping and trunk twisting; symptoms respond to drugs used in human patients | Selective degeneration of dorsal striatal cholinergic interneurons | Predictive validity Time course of motor abnormalities mimics that of human patients | Overexpression: Prevention of abnormal limb clasping and twisting movements Prevention of cholinergic interneuron degeneration |

field, appears normal (*Figure 1—figure supplement 3*). The cerebral cortex is significantly thinner in Emx1(A+B)-CKO than in Emx1-CKO mice (64.8% vs. 10.4% reduction compared to littermate controls, respectively; *Figure 1A,C*). Overall Emx1(A)-CKO CUX1+ and CTIP2+ cell counts (*Figure 1D–1E*) and cortical neuron soma size (*Figure 1—figure supplement 4A*) are unchanged compared to controls. However, restricted counting of CTIP2+ neurons in layer Vb demonstrates a 10.4% reduction in Emx1-CKO mice (*Figure 1—figure supplement 4B*), suggesting a highly specific loss of cells in this sublayer contributes to the reduction in cortical thickness. Emx1(A+B)-CKO mice exhibit significant reductions of CUX1+ and CTIP2+ neurons in sensorimotor cortex (*Figure 1D–E*), demonstrating that torsinB removal increases the susceptibility of multiple cortical neuron populations to torsinA LOF. Consistent with the enhanced neuropathological phenotype, torsinB removal significantly worsens the behavioral phenotype (*Figure 1F*). These mice display reduced postnatal growth (*Figure 1—figure supplement 5A*) and begin to exhibit early lethality beginning in the third postnatal week (*Figure 1—figure supplement 5B*). The humane endpoint for survival and behavioral analyses was P28, precluding studies beyond this age. Considered together with the observation that *Tor1b*$^{-/-}$ mice appear normal, these data suggest that torsinB expression is an essential contributor to neuronal viability on the torsinA null background.

## TorsinB deletion dose-dependently worsens a DYT1 model

To investigate the impact of torsinB reduction in the presence of DYT1 disease mutant (ΔE) torsinA, we examined ΔE torsinA 'selective-knock-in' (SKI) mice (*Liang et al., 2014*). In Emx1-SKI mice (*Emx1-Cre; Tor1a*$^{\Delta E/flx}$), one floxed copy of torsinA is deleted upon *Cre* recombination, leaving isolated

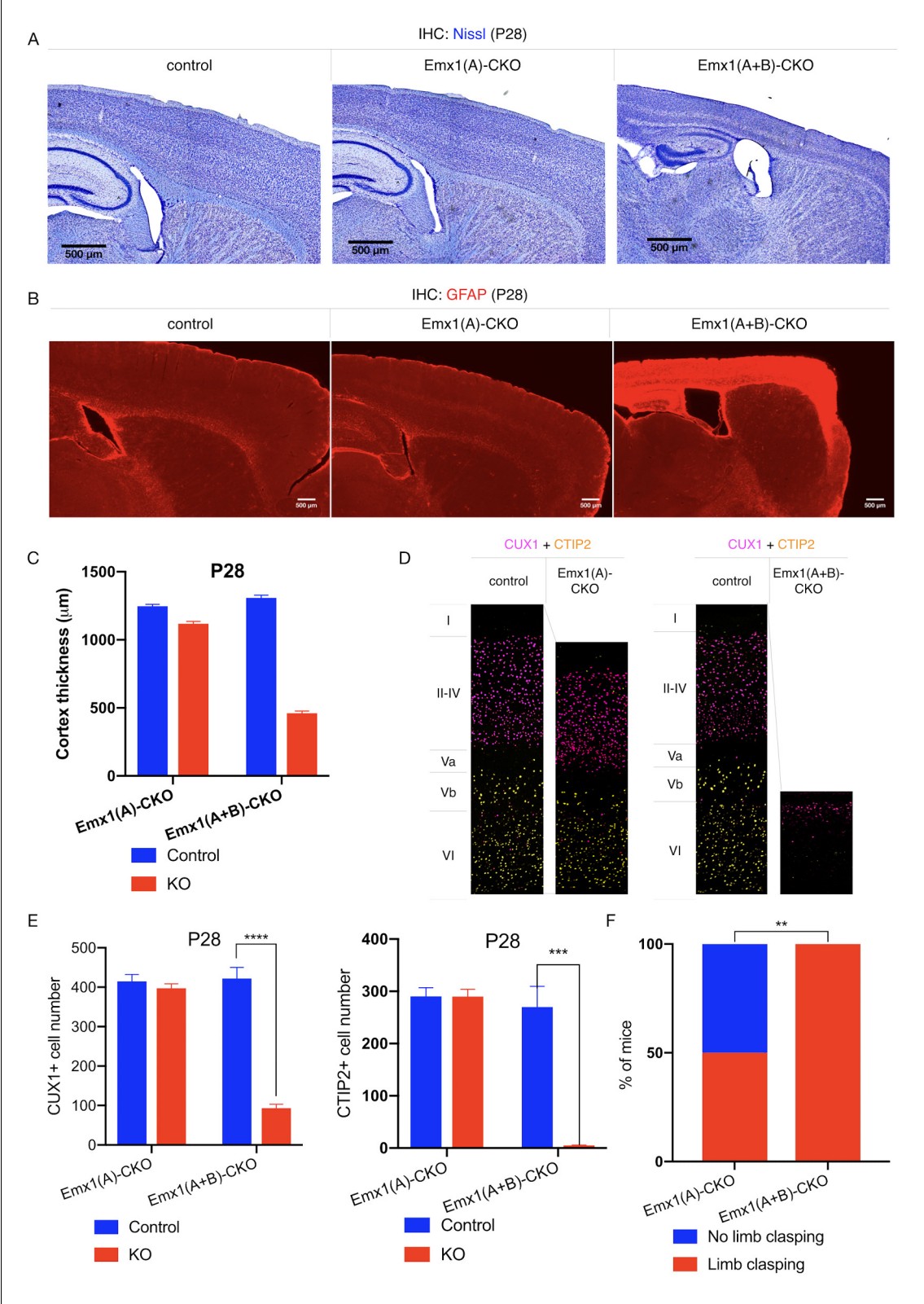

**Figure 1.** TorsinB deletion worsens torsinA-related motor and neuropathological phenotypes. (**A**) Nissl staining of P28 Emx1(A)-CKO and Emx1(A+B)-CKO mice. Emx1(A+B)-CKO mice exhibit significant atrophy of *Cre*-expressing brain regions including cortex (*) and hippocampus (). (**B**) GFAP staining of P28 Emx1(A)-CKO and Emx1(A+B)-CKO brains. Emx1(A+B)-CKO mice exhibit severe reactive gliosis in *Cre*-expressing regions, including the cerebral cortex and the hippocampus. (**C**) Cortical thickness of P28 Emx1(A)-CKO and Emx1(A+B)-CKO mice. Cortical thickness is reduced by 10.4% in Emx1(A)-

*Figure 1 continued on next page*

Figure 1 continued

CKO mice (unpaired t-test $t_{16}$ = 5.834, p<0.0001; control n = 9, Emx1(A)-CKO n = 9). Cortical thickness is reduced by 64.8% in Emx1-dCKO (unpaired t-test $t_{11}$ = 30.16, p<0.0001; control n = 4, Emx1(A+B)-CKO n = 9). (D) Representative images from CUX1 and CTIP2 stained cerebral cortex of Emx1(A)-CKO and Emx1(A+B)-CKO mice and their respective littermate controls. (E) CUX1 and CTIP2 counts in P28 Emx1(A)-CKO and Emx1(A+B)-CKO mice. CUX1+ neurons are not significantly reduced in Emx1(A)-CKO mice (unpaired t-test $t_{16}$ = 0.8469, p=0.4095; control n = 9, Emx1(A)-CKO n = 9). CUX1+ cells are significantly reduced in Emx1(A+B)-CKO mice (77.0% reduction; unpaired t-test $t_8$ = 12.86, p<0.0001, control n = 4, Emx1(A+B)-CKO n = 6). CTIP2+ neurons are not significantly reduced in Emx1(A)-CKO sensorimotor cortex (unpaired t-test $t_{16}$ = 0.02552, p=0.98; control n = 9; Emx1(A)-CKO n = 9). CTIP2+ neurons are significantly reduced in Emx1(A+B)-CKO sensorimotor cortex (98.6%; unpaired t-test $t_7$ = 7.636, p=0.0001, control n = 4, Emx1(A+B)-CKO n = 5). (F) Proportion of Emx1(A)-CKO and Emx1(A+B)-CKO mice exhibiting limb clasping during tail suspension. A significantly greater proportion of Emx1(A+B)-CKO compared to Emx1(A)-CKO mice exhibit limb clasping during tail suspension (Chi square test $\chi^2$ = 9.1, p=0.0026; Emx1(A)-CKO n = 12, Emx1(A+B)-CKO n = 14).

The online version of this article includes the following source data and figure supplement(s) for figure 1:

**Source data 1.** Behavioral and histological data on Emx1(A)-CKO and Emx1(A+B)-CKO mice.
**Figure supplement 1.** TorsinB null mice exhibit no apparent organismal or neuropathological phenotypes.
**Figure supplement 1—source data 1.** Data on histological analysis of torsinB KO mice.
**Figure supplement 2.** Emx1(A+B)-CKO mice do not exhibit neuropathological abnormalities at birth.
**Figure supplement 2—source data 1.** Data on P0 characterization of Emx1(A+B)-CKO mouse brains.
**Figure supplement 3.** Emx1(A+B)-CKO neuropathological abnormalities that emerge postnatally are restricted to the forebrain.
**Figure supplement 4.** Further assessment of cortical neurons in Emx1(A+B)-CKO mice.
**Figure supplement 4—source data 1.** Information pertaining to further characterization of cortical neurons in Emx1(A)-CKO mice.
**Figure supplement 5.** Emx1(A+B)-CKO growth and survival curves.
**Figure supplement 5—source data 1.** Growth and survival data on Emx1(A+B)-CKO mice.

expression of ΔE disease mutant torsinA ($Tor1a^{\Delta E/-}$) in $Emx1$-$Cre$ expressing excitatory forebrain neurons (*Table 1*, *Figure 2A*). We performed a torsinB gene dosage study on Emx1-SKI mice assessing mutants heterozygous (Emx1-SKI;$Tor1b^{+/-}$) or homozygous (Emx1-SKI;$Tor1b^{-/-}$) for a $Tor1b$ null allele (*Figure 2A*, *Figure 2—figure supplement 1A*).

The cortical thickness of Emx1-SKI mice does not differ significantly from littermate controls and these mice have normal numbers of CUX1+ and CTIP2+ cells (*Figure 2B–D*). Ablation of a single torsinB allele (Emx1-SKI;$Tor1b^{+/-}$) significantly reduces cortical thickness but not CUX1+ or CTIP2+ neuron counts (*Figure 2B*, *Figure 2—figure supplement 1C*). The cortical thickness of mice with complete loss of torsinB (Emx1-SKI;$Tor1b^{-/-}$) is normal at birth (*Figure 2—figure supplement 1B*), but by P28 is dramatically reduced to an extent further than in Emx1-SKI;$Tor1b^{+/-}$ mice (*Figure 2B*). At P28, Emx1-SKI;$Tor1b^{-/-}$ mice also exhibit a nonsignificant reduction of CUX1+ neurons (60.9%, p=0.0766) and a significant reduction of CTIP2+ neurons (*Figure 2C–D*). These data highlight that torsinB reduction dose dependently worsens ΔE torsinA LOF-mediated phenotypes. Consistent with a dose-dependent worsening of neuropathology, removal of a single torsinB allele (Emx1-SKI;$Tor1b^{+/-}$) significantly worsened the behavioral phenotype (*Figure 2E*).

## A novel *Cre*-dependent torsinB overexpression allele

Having established the critical nature of endogenous torsinB expression in the context of torsinA LOF, and the developmental- and dose-dependent effects of torsinB expression level, we next explored whether enhancing torsinB levels could suppress or prevent torsinA LOF phenotypes. We generated a torsinB overexpression (B-OE) allele by knocking the $Tor1b$ cDNA into the ROSA26 locus (*Figure 3A*). The $Tor1b$ cDNA is preceded by a floxed 'stop' cassette, rendering this allele *Cre* dependent. This design allows for *Cre* activation of torsinB overexpression in the same spatiotemporal pattern as *Cre* deletion of the floxed $Tor1a$ allele.

We validated the B-OE mouse line by measuring levels of torsinB in brain lysates from *Nestin-Cre* mice, in which the Cre recombinase is active in the cells that give rise to the entire nervous system (*Tronche et al., 1999*). As expected, torsinB is selectively overexpressed only in mice harboring both the *Cre* and B-OE alleles (*Figure 3B–C*). This analysis demonstrates that the B-OE allele supports marked overexpression of torsinB, likely in part because of the included woodchuck hepatitis virus posttranscriptional regulatory element. *Nestin-Cre*;B-OE mice exhibit normal torsinB expression in liver (a *Cre* negative tissue), demonstrating that overexpression is selective and *Cre*-dependent (*Figure 3—figure supplement 1A–B*). Quantification of control torsinB levels and diluted torsinB

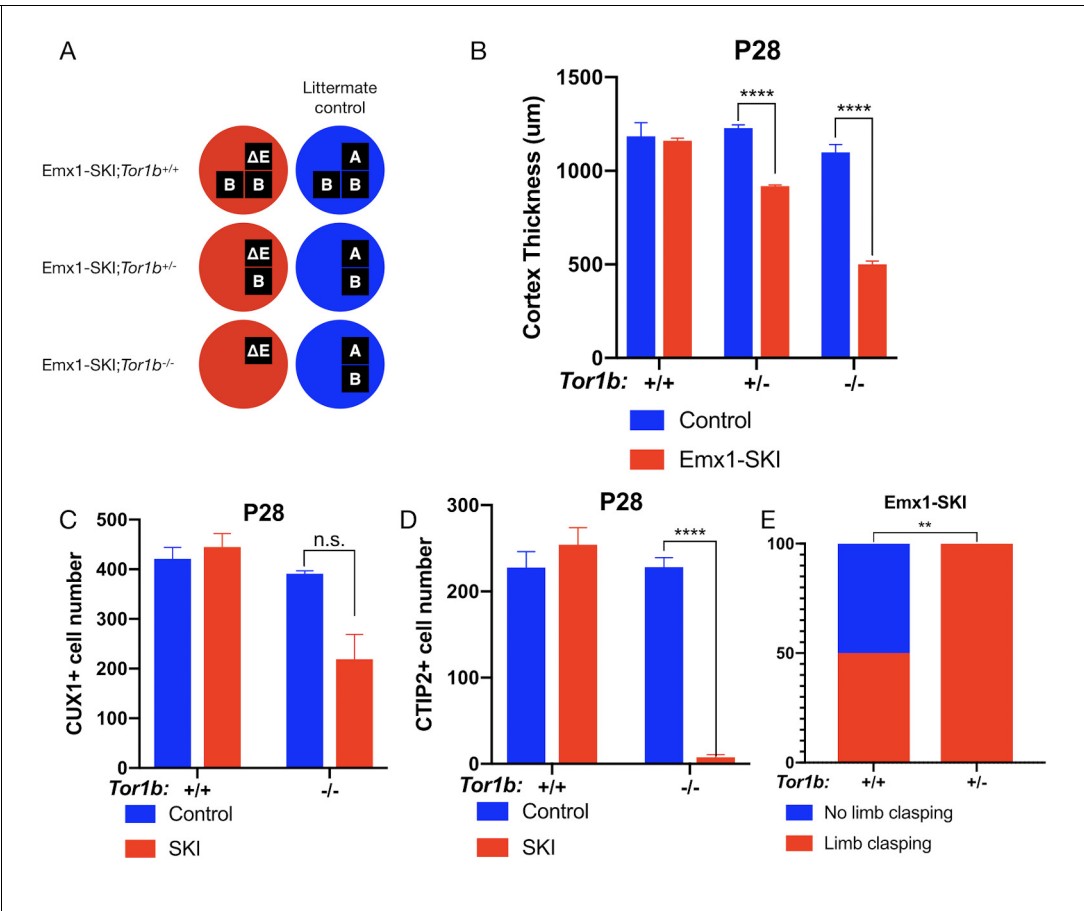

**Figure 2.** TorsinB dose-dependently worsens a DYT1 model. (A) Illustration of examined genotypes. Each row of boxes within the circles denote the presence or absence of *Tor1a* (top row) or *Tor1b* (bottom row) alleles following Cre recombination. The presence of a letter ('A' 'ΔE' or 'B') indicates an intact allele, whereas the absence of a letter indicates a deleted allele. (B) Cortical thickness of P28 Emx1-SKI;*Tor1b*$^{+/+}$, Emx1-SKI;*Tor1b*$^{+/-}$, and Emx1-SKI;*Tor1b*$^{-/-}$ mice. TorsinB loss reduces cortical thickness in Emx1-SKI mice in a dose-dependent manner (2% reduction in Emx1-SKI;*Tor1b*$^{+/+}$, 25.2% reduction in Emx1-SKI;*Tor1b*$^{+/-}$, and 54.5% decrease in Emx1-SKI;*Tor1b*$^{-/-}$; two way ANOVA main effect of background genotype $F_{1,13}$ = 128.3, p<0.0001, age $F_{2,13}$ = 62.65, p<0.0001, and interaction $F_{2,13}$ = 36.98, p<0.0001; Sidak's multiple comparisons test p=0.9323 for *Tor1b*$^{+/+}$, p<0.0001 for *Tor1b*$^{+/-}$, p<0.0001 for Tor1b$^{-/-}$; *Tor1b*$^{+/+}$ control n = 3; Emx1-SKI n = 5, *Tor1b*$^{+/-}$ control n=3, Emx1-SKI n = 3, *Tor1b*$^{-/-}$ control n=2, Emx1-SKI n = 3). (C) CUX1+ cell counts in sensorimotor cortex of P28 Emx1-SKI;*Tor1b*$^{+/+}$ and Emx1-SKI;*Tor1b*$^{-/-}$ mice. There is no significant reduction in the number of CUX1+ cells in Emx1-SKI;*Tor1b*$^{+/+}$ sensorimotor cortex (unpaired t-test $t_{16}$ = 0.8469, p=0.4095; control n = 3, SKI n = 5). Simultaneous deletion of two torsinB alleles reduces CUX1+ cell counts by 44.0% (unpaired t-test $t_3$ = 2.655, p=0.0766; control n = 2, SKI n = 3) though this reduction does not reach statistical significance. (D) CTIP2+ cell counts in sensorimotor cortex of P28 Emx1-SKI;*Tor1b*$^{+/+}$ and Emx1-SKI;*Tor1b*$^{-/-}$ mice. CTIP2+ neuronal cell counts are not reduced in Emx1-SKI;*Tor1b*$^{+/+}$ mice (unpaired t-test $t_6$ = 0.8844, p=0.4105; control n = 3, SKI n = 5). CTIP2+ cell counts in Emx1-SKI;*Tor1b*$^{-/-}$ are significantly reduced by 96.6% (unpaired t-test $t_3$ = 24.44, p<0.0001; control n = 2, SKI n = 3). (E) Proportion of Emx1-SKI;*Tor1b*$^{+/+}$ and Emx1-SKI;*Tor1b*$^{+/-}$ mice exhibiting limb clasping during tail suspension. A significantly greater proportion of Emx1-SKI;*Tor1b*$^{+/-}$ mice exhibit limb clasping during tail suspension compared to Emx1-SKI;*Tor1b*$^{+/+}$ (Chi square test $\chi^2$ = 7.441, p=0.0064; Emx1-SKI;*Tor1b*$^{+/+}$ n = 12, Emx1-SKI;*Tor1b*$^{+/-}$ n = 11).

The online version of this article includes the following source data and figure supplement(s) for figure 2:

**Source data 1.** Raw data on histological and behavioral characterization of Emx1-SKI mice with two, one, and zero intact torsinB alleles.

**Figure supplement 1.** Novel allele design and histological analysis of Emx1-SKI;*Tor1b*$^{-/-}$ mice and Emx1-SKI;*Tor1b*$^{+/-}$ mice.

**Figure supplement 1—source data 1.** Further histological characterization of Emx1-SKI mice with varying amounts of torsinB.

overexpression levels indicates a ~30 fold increase of torsinB expression (*Figure 3C*). We next tested whether torsinB level impacts torsinA expression, as past studies suggest that torsinA and torsinB have reciprocal timing and tissue patterns of expression (*Kim et al., 2010*; *Tanabe et al., 2016*). TorsinB overexpression did not significantly alter torsinA expression in brain lysates, indicating that

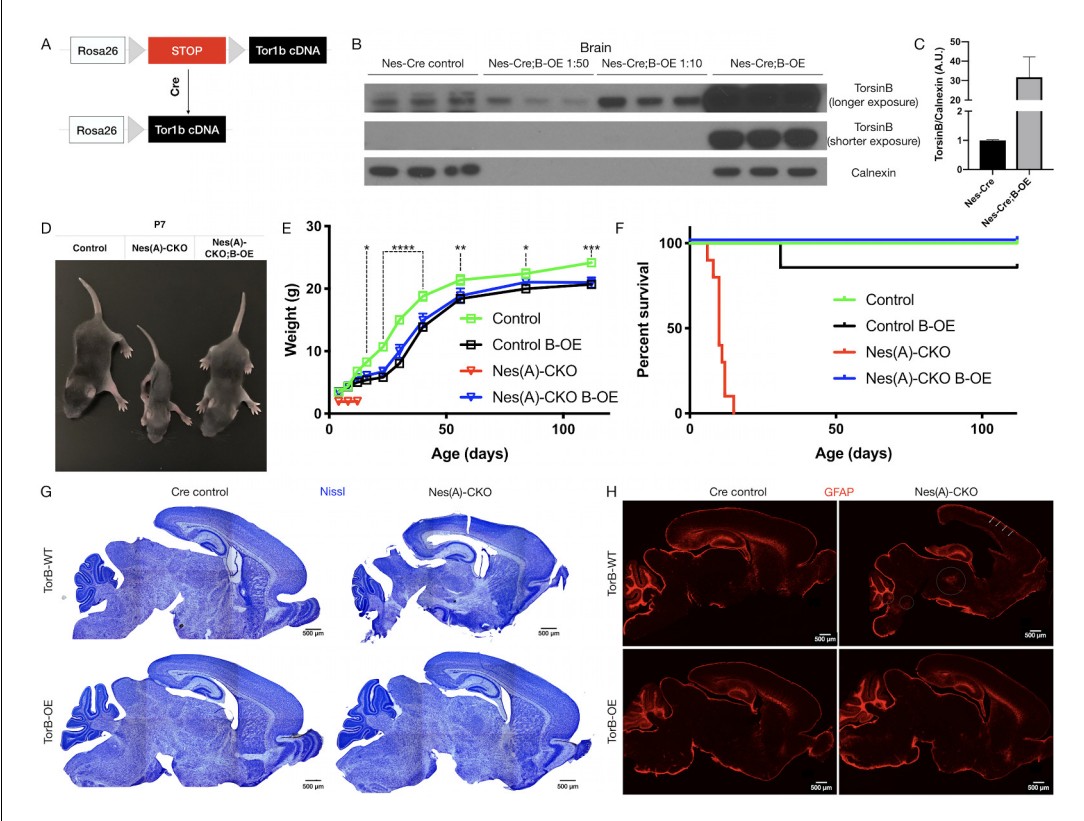

**Figure 3.** TorsinB augmentation eliminated all phenotypes in CNS conditional torsinA null mice. (**A**) Cartoon illustration of ROSA26 locus engineered to express torsinB. Top: In absence of *Cre*, torsinB expression is prevented by a floxed 'STOP' cassette. Bottom: TorsinB is expressed following *Cre* deletion of the floxed 'STOP' cassette. (**B**) Western blot analysis of whole brain lysates probed with an anti-torsinB antibody. Mice expressing both the *Nestin-Cre* and B-OE alleles exhibit *Cre*-dependent overexpression of torsinB. (**C**) Quantification of western blots of whole brain lysates from *Nestin-Cre* and *Nestin-Cre*;B-OE littermates probed with anti-torsinB antibody. The B-OE allele causes significant torsinB overexpression (unpaired t-test $t_4$ = 2.947, p=0.0421; *Nestin-Cre* n = 3, *Nestin-Cre*;B-OE n = 3). Quantification was performed by comparing torsinB expression in *Cre* control samples to that of *Nestin-Cre*;B-OE samples diluted 1:50. (**D**) Image of *Cre* control, Nes(A)-CKO, and Nes(A)-CKO;B-OE littermates at P7 (in order left to right). TorsinB overexpression restores postnatal growth. (**E**) Growth curves of *Cre* control, *Cre* control;B-OE, N(esA)-CKO, and Nes(A)-CKO;B-OE mice. TorsinB overexpression almost entirely restores growth in N-CKO mice (two-way repeated measures ANOVA main effect of genotype $F_{2, 17}$ = 14.16, p=0.0002, main effect of age $F_{9, 153}$ = 775.7 p<0.0001, interaction $F_{18, 153}$ = 6.727, p<0.0001; asterisks denote the following p-values from Tukey's multiple comparisons tests: *=p < 0.05, **=p < 0.01, **=p < 0.001, ****=p < 0.0001; *Cre* control n = 7, *Cre* control;B-OE n = 6, N(A)-CKO;B-OE n = 7). (**F**) Survival curves of *Cre* control, *Cre* control;B-OE, Nes(A)-CKO, and Nes(A)-CKO;B-OE mice. TorsinB overexpression eliminates lethality in N-CKO mice (Gehan-Breslow-Wilcoxon method $\chi^2$ = 36.16, p<0.0001; *Cre* control n = 7, *Cre* control;B-OE n = 7, Nes(A)-CKO n = 10, Nes(A)-CKO;B-OE n = 7). (**G**) Nissl staining of P8 brains from *Cre* control, *Cre* control;B-OE, Nes(A)-CKO, and Nes(A)-CKO;B-OE mice. TorsinB overexpression eliminates the morphological defects characteristic of Nes(A)-CKO mice. (**H**) GFAP staining of P8 brains from *Cre* control, *Cre* control;B-OE, Nes(A)-CKO, and Nes(A)-CKO;B-OE mice. GFAP immunostaining illustrates the characteristic gliotic changes in Nes(A)-CKO cortex (arrows), thalamus, deep cerebellar nuclei, and hindbrain (circles). TorsinB overexpression eliminates gliotic changes in Nes(A)-CKO mice.

The online version of this article includes the following source data and figure supplement(s) for figure 3:

**Source data 1.** Biochemical and organismal characterization of Nes(A)-CKO;B-OE mice.
**Figure supplement 1.** Quantification of torsinB in *Nestin-Cre*;B-OE liver tissue.
**Figure supplement 1—source data 1.** Information pertaining to torsinA expression in torsinB overexpression brain tissue.
**Figure supplement 2.** Analysis of torsinA expression in *Nestin-Cre*;B-OE brain tissue.
**Figure supplement 2—source data 1.** Information on liver torsinB expression levels.
**Figure supplement 3.** TorsinB overexpression prevents overtly abnormal postures and gross disruption of brain morphology in N-CKO mice.
**Figure supplement 3—source data 1.** Behavioral and brain morphological characterization of Nes(A)-CKO;B-OE mice.

there is not a direct regulatory relationship between levels of these proteins (*Figure 3—figure supplement 2A–B*). These findings confirm that this line functions as designed. We employed this line to explore whether torsinB overexpression can rescue torsinA LOF phenotypes.

## TorsinB augmentation eliminates all phenotypes in CNS conditional torsinA null mice

Prior work demonstrates that torsinB overexpression significantly suppresses a torsinA LOF cellular phenotype in vitro. To determine if torsinB overexpression can rescue torsinA LOF behavioral and neuropathological phenotypes in vivo, we ingressed the *Cre*-dependent B-OE allele onto the *Nestin-Cre* torsinA conditional knockout (Nes(A)-CKO) background. Nes(A)-CKO mice exhibit early lethality, lack of postnatal weight gain, overtly abnormal twisting movements, and gliosis in multiple sensori-motor brain regions (*Table 1*; *Liang et al., 2014*), robust phenotypes that can be harnessed to determine if exogenous torsinB can compensate for torsinA LOF. We analyzed four genotypes: *Nestin-Cre; Tor1a*$^{flx/+}$ (Cre control), *Nestin-Cre; Tor1a*$^{flx/+}$; B-OE (*Cre* control;B-OE), *Nestin-Cre; Tor1a*$^{flx/-}$ (Nes(A)-CKO), and *Nestin-Cre; Tor1a*$^{flx/-}$; B-OE (Nes(A)-CKO;B-OE). Consistent with prior work, Nes(A)-CKO mice weighed less than their littermates controls and exhibited 100% lethality by P15 (*Figure 3C–E*; *Liang et al., 2014*). As described previously, Nes(A)-CKO mice demonstrated overt abnormal twisting movements and stiff postures (*Liang et al., 2014*). TorsinB overexpression completely eliminated these abnormal movements (*Figure 3—figure supplement 3A*). Strikingly, torsinB overexpression also eliminated Nes(A)-CKO lethality and significantly rescued postnatal growth (*Figure 3C–E*). The widespread non-physiologic levels of torsinB produced by the B-OE allele impaired bodyweight in *Nestin-Cre* control mice (*Figure 3D*; comparing *Cre* control and *Cre* control;B-OE), but did not induce overt motor abnormalities or reduce survival (Gehan-Breslow-Wilcoxon method $\chi^2 = 1$, p=0.3173; *Cre* control n = 7, *Cre* control;B-OE n = 7). Nissl staining at P8 (prior to lethality) demonstrated the expected Nes(A)-CKO neuropathology, including Nes(A)-CKO brains exhibited enlarged lateral ventricles and a reduction in overall brain size compared to *Cre* controls (*Figure 3G*, *Figure 3—figure supplement 3B*). Brain morphology and size appeared normal in both *Cre* control;B-OE and Nes(A)-CKO;B-OE mice at the same age (*Figure 3F*, *Figure 3—figure supplement 3B*). Consistent with prior work, the brains of Nes(A)-CKO mice exhibited GFAP staining reflecting gliosis in the cerebral cortex, thalamus, brainstem, and deep cerebellar nuclei (*Figure 3G*; (*Liang et al., 2014*). In striking contrast, torsinB overexpression prevented these gliotic changes (*Figure 3G*).

## TorsinB overexpression rescues ΔE torsinA phenotypes

The rescue of the Nes(A)-CKO model demonstrates that torsinB can significantly ameliorate torsinA LOF phenotypes, even when torsinA is absent from the entire central nervous system. However, DYT1 dystonia is dominantly inherited demonstrating that heterozygous ΔE mutation is pathogenic in humans (*Ozelius et al., 1997*). TorsinA is proposed to function in oligomeric complexes (*Chase et al., 2017*; *Laudermilch and Schlieker, 2016*; *Demircioglu et al., 2019*), suggesting that the ΔE torsinA allele may exert dominant negative effects (*Torres et al., 2004*; *Goodchild and Dauer, 2004*). This raises the question of whether torsinB overexpression can rescue torsinA LOF phenotypes in the presence of ΔE torsinA.

We employed the *Nestin-Cre* selective-knock-in model (Nes-SKI), in which *Cre* recombination results in isolated expression of ΔE torsinA (Tor1a$^{\Delta E/-}$) in the entire central nervous system (*Table 1*; *Liang et al., 2014*). These mice exhibit reduced postnatal growth and reactive gliosis and neurodegeneration in a similar but milder pattern as in Nes(A)-CKO (*Liang et al., 2014*). We combined this model with the B-OE allele and analyzed four genotypes: *Cre* control (*Nestin-Cre; Tor1a*$^{flx/+}$), *Cre* control;B-OE (*Nestin-Cre; Tor1a*$^{flx/+}$; B-OE), Nes-SKI (Nestin-Cre *Tor1a*$^{\Delta E/flx}$), and Nes-SKI;B-OE (*Nestin-Cre; Tor1a*$^{\Delta E/flx}$; B-OE). We hypothesized that torsinB overexpression would rescue mouse phenotypes in the presence of ΔE torsinA.

TorsinB overexpression restored postnatal growth and prevented neurodegeneration in Nes-SKI mice (*Figure 4*). Nes-SKI;B-OE mice gained significantly more weight than Nes-SKI mice. While Nes-SKI mice weighed significantly less than *Cre* control littermates at all ages measured (Tukey's multiple comparisons test, p<0.0001 at P8, p<0.0001 at P15, p=0.0001 at P21, p=0.0002 at P28, p=0.0056 at P56; *Figure 4A*), torsinB overexpression partially rescued postnatal growth from P8-P28

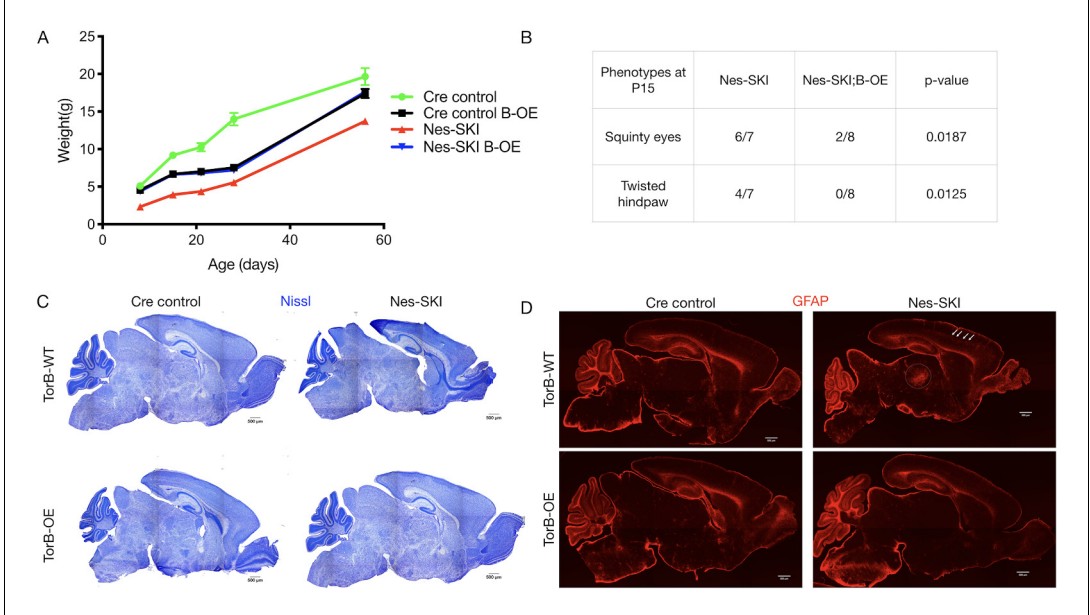

**Figure 4.** TorsinB overexpression rescues ΔE torsinA phenotypes. TorsinB overexpression prevents striatal cholinergic interneuron degeneration and dystonic-like movements. (**A**) Growth curves of *Cre* control, *Cre* control;B-OE, Nes-SKI, and Nes-SKI;B-OE mice. Nes-SKI mice exhibited reduced postnatal growth, which was partially rescued by torsinB overexpression (two-way repeated measures ANOVA main effect of genotype $F_{3,\,26}$ = 60.65, p<0.0001, main effect of age $F_{1.828,\,47.53}$ = 794.8 p<0.0001, interaction $F_{12,\,104}$ = 9.831, p<0.0001; *Cre* control n = 7, *Cre* control;B-OE n = 8, Nes-SKI n = 7, Nes-SKI;B-OE n = 8). (**B**) Table of overtly abnormal postural and developmental phenotypes displayed by Nes-SKI and Nes-SKI;B-OE mice at P15. TorsinB overexpression reduces phenotypes that are prevalent in Nes-SKI mice (squinty eyes: Chi square test χ2 = 5.529, p=0.0187; twisted hindpaw: Chi square test χ2 = 6.234, p=0.0125). (**C**) Nissl staining of P10 brains from *Cre* control, *Cre* control;B-OE, Nes-SKI, and Nes-SKI;B-OE mice. TorsinB overexpression eliminates the morphological defects characteristic of Nes-SKI mice. (**D**) GFAP staining of P10 brains from *Cre* control, *Cre* control;B-OE, Nes-SKI, and Nes-SKI;B-OE mice. GFAP immunostaining illustrates the characteristic gliotic changes in Nes-SKI, including in cortex (arrows) and thalamus (circle). TorsinB overexpression eliminates gliotic changes in Nes-SKI mice.

The online version of this article includes the following source data for figure 4:

**Source data 1.** Information characterizing Nes-SKI;B-OE mice.

and fully restored normal weight by P56 in Nes-SKI;B-OE mice (Tukey's multiple comparisons test, Nes-SKI;B-OE vs. *Cre* control, p=0.3972) (*Figure 4A*). *Cre* control;B-OE mice exhibited reduced weight compared to *Cre* controls with normal endogenous torsinB before P56, and weighed significantly more than Nes-SKI mice (*Figure 4A*). This intermediate weight was similar to the *Cre* control; B-OE mice in the N(A)-CKO study (*Figure 3D*). Nes-SKI mice exhibit abnormal postural phenotypes at rest (*Liang et al., 2014*). TorsinB overexpression significantly reduced the prevalence of squinty eyes and twisted hindpaws at P15 (*Figure 4B*). As expected, Nes-SKI mice had gross morphologic disruption of brain structure (*Figure 4C*) and gliosis in multiple sensorimotor regions (*Figure 4D*). TorsinB overexpression abolished both of these neuropathological phenotypes (*Figure 4C–D*). These data demonstrate that the presence of ΔE torsinA does not blunt the ability of torsinB overexpression to rescue torsinA LOF phenotypes.

## TorsinB overexpression prevents striatal cholinergic interneuron degeneration and dystonic-like movements

Conditional deletion of torsinA from the forebrain using *Dlx5/6-Cre* ('Dlx(A)-CKO') produces a robust DYT1 model with high face and predictive validity (*Table 1*; *Pappas et al., 2015*). These mutants develop abnormal limb and trunk twisting movements as juveniles that are responsive to anti-muscarinic drugs such as trihexyphenidyl, mimicking key features of the human phenotype (*Greene et al., 1995*; *Bressman et al., 2000*; *Cloud and Jinnah, 2010*). Coincident with the emergence of these movements, these animals exhibit a highly selective loss of dorsal striatal cholinergic interneurons (ChIs), cells that have been implicated in dystonia pathophysiology in electrophysiological studies (*Eskow Jaunarajs et al., 2015*; *Pappas et al., 2015*; *Scarduzio et al., 2017*). The disease relevant

time course of motor features, clear endpoints, and predictive validity of this model provide a critical system for evaluating the potential benefit of torsinB overexpression in a torsinA LOF pathophysiological in vivo setting. To explore the ability of the B-OE allele to suppress or prevent phenotypes in this model, we first confirmed that this allele functions as expected with the *Dlx5/6-Cre* transgene and assessed for any behavioral effects caused by overexpressing torsinB in this Cre field. Mice carrying the *Dlx5/6-Cre* and the B-OE allele selectively overexpressed torsinB in the forebrain in a *Cre*-dependent manner (*Figure 5—figure supplement 1A*). TorsinB overexpression in the *Dlx5/6-Cre* field (*Dlx5/6-Cre*;B-OE) caused no abnormalities of postnatal growth (*Figure 5—figure supplement 1B*) or survival (all mice survived; data not shown) compared to mice only expressing *Dlx5/6-Cre*. Further, *Dlx5/6-Cre*;B-OE mice exhibited normal juvenile reflexes, including negative geotaxis, surface righting, and forelimb hang (*Figure 5—figure supplement 1C*).

Having confirmed the proper functioning and lack of toxicity of the B-OE allele when activated in the *Dlx5/6-Cre* field, we next examined the effect of torsinB overexpression on Dlx-CKO motor and neuropathological phenotypes. We examined four genotypes: *Dlx5/6-Cre; Tor1a$^{flx/+}$* (Cre control), *Dlx5/6-Cre; Tor1a$^{flx/+}$*; B-OE (Cre control;B-OE), *Dlx5/6-Cre; Tor1a$^{flx/-}$* (Dlx(A)-CKO), and *Dlx5/6-Cre; Tor1a$^{flx/-}$*; B-OE (Dlx(A)-CKO;B-OE). Overexpression of torsinB significantly suppressed abnormal twisting movements in Dlx(A)-CKO mice. We recorded 1 min tail suspension videos at age P70, and two observers blind to genotype scored each video for duration of abnormal movements. Dlx(A)-CKO mice exhibited limb clasping for an average of 36.9 ± 5.3 s (mean ± SEM). TorsinB overexpression (Dlx(A)-CKO;B-OE) significantly reduced the time spent clasping (2.5 ± 1. 1 s, mean ± SEM) by over 10-fold (*Figure 5A*). Notably, the duration of clasping in Dlx(A)-CKO;B-OE did not differ significantly from their littermate controls. We also assessed the mice for the presence of trunk twisting during tail suspension. This phenotype occurred in most (9 of 11) Dlx(A)-CKO mice but was completely eliminated by the B-OE allele (0/15 trunk twisting, p<0.0001 compared to Dlx(A)-CKO; *Figure 5B*). Neither *Cre* control nor *Cre* control;B-OE mice exhibited trunk twisting while suspended (data not shown). These data demonstrate the ability of torsinB to suppress motor phenotypes in a validated DYT1 model.

We analyzed the brains of all four genotypes to identify potential neuropathological correlates of the B-OE-mediated behavioral rescue. TorsinB overexpression did not change cortical thickness, striatal volume, striatal Nissl+ small and medium cell number, or striatal GFAP immunoreactivity in Dlx(A)-CKO or *Cre* control mice (*Figure 5—figure supplement 2A–D*). TorsinB overexpression completely prevented the loss of dorsal striatal ChIs that have been linked to motor and electrophysiologic abnormalities in dystonia mouse models (*Maltese et al., 2014*; *Pappas et al., 2015*; *Eskow Jaunarajs et al., 2015*; *Scarduzio et al., 2017*; *Pappas et al., 2018*; *Eskow Jaunarajs et al., 2019*; *Richter et al., 2019*). Analysis using unbiased stereology demonstrated that Dlx(A)-CKO mice had 33.5% fewer ChIs compared to *Cre* controls, whereas the B-OE allele completely prevented this loss (*Figure 5C–D*). Dlx(A)-CKO;B-OE mice did not statistically differ from *Cre* controls (mean ± SEM of 17039 ± 534 vs. 17252 ± 432 cells, respectively). Considered together, these data demonstrate the ability of torsinB to potently suppress DYT1-related motor and neuropathological phenotypes.

## Discussion

Our studies establish torsinB as a bidirectional modulator of torsinA dystonia-related motor phenotypes. We demonstrate that reductions of torsinB cause a dose-dependent worsening of neuropathological and motor abnormalities in multiple DYT1 models, including one containing the pathogenic ΔE disease mutation. In contrast, torsinB supplementation essentially eliminates the phenotypes in these models. These data suggest that torsinB may be an effective therapeutic target in DYT1 dystonia.

These data advance understanding of the relationship of torsinB to DYT1 pathogenesis. Prior studies established a role for torsinB in the tissue selectivity and timing of torsinA-related cell biological phenotype of NE budding (*Kim et al., 2010*; *Tanabe et al., 2016*). These new data provide proof-of-principle evidence that torsinB enhancing therapies may modify disease course. Increasing the expression or function of torsinA itself is an alternative approach. However, torsinA-targeted therapies would similarly increase levels of mutant ΔE torsinA, potentially worsening disease severity via a dominant negative mechanism. TorsinB-targeted therapies, in contrast, should bypass this risk. The nonphysiologic torsinB overexpression of the B-OE allele in the *Nestin-Cre* field caused reduced

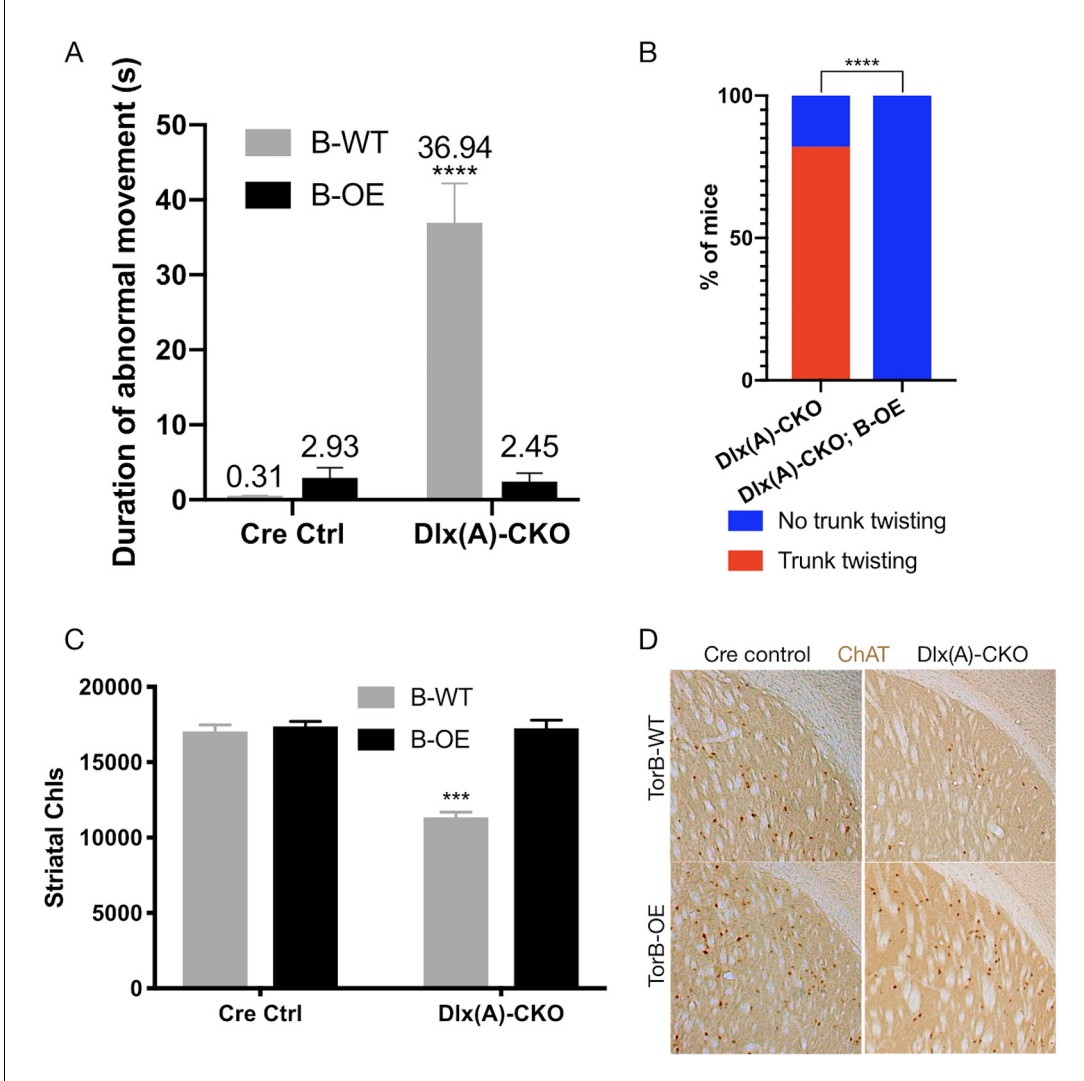

**Figure 5.** TorsinB overexpression prevents striatal cholinergic interneuron degeneration and dystonic-like movements. (**A**) Duration of abnormal movements during one minute of tail suspension in P70 *Cre* control, *Cre* control;B-OE, Dlx(A)-CKO, and Dlx(A)-CKO;B-OE mice. TorsinB overexpression significantly reduces severity of limb clasping (two-way ANOVA main effect of genotype $F_{1,44}$ = 47.45, p<0.0001, torsinB level $F_{1,44}$ = 36.86, p<0.0001, and interaction $F_{1,44}$ = 49.96, p<0.0001; *Cre* control n = 10, *Cre* control;B-OE n = 12, Dlx(A)-CKO n = 11, Dlx(A)-CKO;B-OE n = 15). (**B**) Prevalence of trunk twisting in Dlx(A)-CKO and Dlx(A)-CKO;B-OE mice. TorsinB overexpression significantly reduces prevalence of trunk twisting (Chi square test $\chi^2$ = 18.77, p<0.0001; Dlx(A)-CKO n = 11, Dlx(A)-CKO; B-OE n = 15). (**C**) Stereologic counts of striatal cholinergic interneurons in P70 *Cre* control, *Cre* control;B-OE, Dlx(A)-CKO, and Dlx(A)-CKO;B-OE brains. TorsinB overexpression prevents ChI degeneration characteristic of Dlx(A)-CKO mice (two-way ANOVA main effect of genotype $F_{1,22}$ = 52.45, p<0.0001, ROSA-*Tor1b* allele $F_{1,22}$ = 45.54, p<0.0001, and interaction $F_{1,22}$ = 41.90, p<0.0001; *Cre* control n = 6, *Cre* control;B-OE n = 6, Dlx(A)-CKO n = 7, Dlx(A)-CKO;B-OE n = 7). (**D**) Representative images of P70 striatum immunostained with antibody to ChAT. ChAT+ cell density is reduced in Dlx(A)-CKO striatum while it appears normal in Dlx(A)-CKO;B-OE striatum.

The online version of this article includes the following source data and figure supplement(s) for figure 5:

**Source data 1.** Information on behavioral and histological characterization of Dlx(A)-CKO;B-OE mice.

**Figure supplement 1.** Dlx-Cre;B-OE mice exhibit no apparent motor or organismal phenotype.

**Figure supplement 1—source data 1.** Infomation pertaining to growth and preweaning reflexes of Dlx-Cre;B-OE mice.

**Figure supplement 2.** Dlx-Cre;B-OE mice exhibit no apparent neuropathological phenotype.

**Figure supplement 2—source data 1.** Information on neuropathological characterization of Dlx(A)-CKO;B-OE mice.

growth in *Cre* control mice, highlighting potential adverse consequences of this approach. The B-OE mice grossly overexpress torsinB at a level ~30 times higher than controls (*Figure 3B*, *Figure 3—figure supplement 1A*). Considering that removal of a single torsinB allele is sufficient to worsen

behavioral and neuropathological readouts (*Figure 2B and E*), it is likely that more modest overexpression will provide protective effects. Future work will be needed to determine the therapeutic window of torsinB supplementation.

Along these lines, future therapeutic translation of this work requires the identification of safe and effective methods to modulate torsinB. One future direction will be to identify compounds or signaling pathways that impact torsinB expression. Cell culture-based genetic screens to identify torsinB-related pathways, and high-throughput systems to screen compounds that increase torsinB expression are two in vitro approaches that could lead to further preclinical testing. Gene therapies to enhance torsinB expression are also an important strategy to explore, whether through viral-mediated delivery of torsinB itself, or antisense oligonucleotides to stabilize torsinB transcripts or boost its expression. A significant advantage of AAV-mediated expression compared to our mouse genetic approach is that the dose of AAV injection is adjustable, which could help elucidate the therapeutic window of torsinB overexpression.

The interplay between torsinA LOF and torsinB levels is reminiscent of spinal muscular atrophy (SMA), a disease linked to LOF of the SMN1 gene (*Kolb and Kissel, 2015*). The disease course of SMA is characterized by highly variable severity and age of onset, which is caused in large part by differences in copy number of SMN2, a duplicate of SMN1 that can attenuate SMN1 loss (*Arnold et al., 2016*; *Bharucha-Goebel and Kaufmann, 2017*; *Garcia-Lopez et al., 2018*). In our dystonia mouse models, a similar relationship exists where torsinB levels bidirectionally modulate the severity of torsinA LOF, worsening mouse phenotypes when reduced and attenuating them when overexpressed. The effect of differences in torsinB expression on DYT1 penetrance and severity in human subjects is unknown. Further human genetic studies will be required to explore this question. The most direct correlative method would be to compare torsinB expression level in neural tissue samples from subjects with varying symptom severity and to compare manifesting and non-manifesting DYT1 mutation carriers. However, postmortem samples from patients with inherited dystonia are exceedingly rare, especially at the young ages that would be most relevant to predicting disease course. An alternative would be to assess torsinB levels in non-neural samples from living patients, but mouse data demonstrate opposing patterns of torsinA and torsinB expression in CNS and peripheral tissues (*Jungwirth et al., 2010*; *Kim et al., 2010*). A potential alternative approach is to derive neurons from fibroblasts of control subjects, non-manifesting DYT1 mutation carriers, and manifesting carriers of varying severity. Assessment of torsinB levels in these cells could provide hints into the role of torsinB in disease severity or penetrance.

Neurodegeneration is one of the primary readouts in this study, whereas inherited isolated dystonia is commonly thought to be characterized by abnormal function within a structurally normal brain. Postmortem studies are limited, however, and volumetric imaging studies provide conflicting evidence (*Tanabe et al., 2009*; *Paudel et al., 2012*; *den Dunnen, 2013*; *Ramdhani and Simonyan, 2013*). Future studies are required to fully assess potential neuropathological changes in DYT1 dystonia, especially considering that a diverse range of basal ganglia insults cause acquired dystonia (*Pettigrew and Jankovic, 1985*; *Timmermann et al., 2010*; *Bansil et al., 2012*). Considerable evidence suggests that striatal cholinergic dysfunction is a key feature contributing to this disorder (*Pappas et al., 2015*; *Eskow Jaunarajs et al., 2015*; *Deffains and Bergman, 2015*), supporting our focus on these cells. Our findings that torsinB overexpression prevents both striatal ChI degeneration and twisting movements further strengthens the relationship between dysfunction of these cells and DYT1-related abnormal movements.

This study provides further evidence that torsinA-LOF mediated neurodegeneration is cell autonomous. Previous data have demonstrated that in torsinA *Cre* conditional knockout mice, neurodegeneration is selective to *Cre* expressing brain regions (*Liang et al., 2014*; *Weisheit and Dauer, 2015*). Furthermore, Dlx(A)-CKO and cholinergic-selective deletion of torsinA exhibit indistinguishable degree and topography of striatal ChI degeneration (*Pappas et al., 2015*; *Pappas et al., 2018*). Our rescue studies utilize *Cre* recombinase to delete torsinA and activate torsinB overexpression in the same spatiotemporal pattern, demonstrating a cell autonomous rescue of torsinA LOF-mediated neurodegeneration. A related question raised by our results is whether neuronal or glial torsinB overexpression contributes to the to prevention of neurodegeneration. While *Nestin-Cre* expresses in neural progenitor cells that give rise to both neurons and glia (*Tronche et al., 1999*), *Dlx5/6-Cre* is exclusive to forebrain GABAergic and cholinergic neurons (*Monory et al., 2006*), suggesting that neuron-specific torsinB overexpression is sufficient to prevent neurodegeneration.

Our findings advance the understanding of the role of torsinB in DYT1 dystonia and demonstrate that torsinB expression is a bidirectional modifier of torsinA LOF phenotypes. TorsinB overexpression is a potent disease-modifier, reducing both the prevalence and severity of abnormal movements, and fully preventing neuropathological abnormalities. These data provide a strong rationale and identify future directions to continue exploring torsinB as a target for DYT1 dystonia therapeutics.

# Materials and methods

**Key resources table**

| Reagent type (species) or resource | Designation | Source or reference | Identifiers | Additional information |
|---|---|---|---|---|
| Antibody | Rabbit polyclonal α-GFAP | Dako | Cat#Z0334 RRID:AB_10013382 | IHC(1:2000) |
| Antibody | Rabbit polyclonal α-CUX1 | Santa Cruz Biotechnology | Cat#sc-13024 RRID:AB_2261231 | IHC(1:200) |
| Antibody | Rat monoclonal α-CTIP2 | Abcam | Cat#ab18465 RRID:AB_2064130 | IHC(1:500) |
| Antibody | Goat polyclonal α-ChAT | Millipore | Cat#AB144P RRID:AB_90560 | IHC(1:200) |
| Antibody | Donkey α-rabbit IgG, Alexa Fluor 555 conjugated | Invitrogen | Cat#A-31572 RRID:AB_162543 | IHC(1:800) |
| Antibody | Donkey α-rat IgG, Alexa Fluor 488 conjugated | Invitrogen | Cat# A-21208 RRID:AB_141709 | IHC(1:800) |
| Antibody | Donkey α-goat IgG, biotin-SP conjugated | Jackson Immunoresearch | Cat#705-065-147 RRID:AB_2340397 | IHC(1:800) |
| Antibody | Rabbit polyclonal α-torsinA | Abcam | #ab34540 RRID:AB_2240792 | WB(1:10,000) |
| Antibody | Rabbit polyclonal α-torsinB | Kind gift of Schlieker lab | N/A | WB(1:1,000) |
| Antibody | Goat α-rabbit IgG, HRP-linked | Cell Signaling | Cat#7074 RRID:AB_2099233 | WB(1:20,000) |
| Antibody | Rabbit polyclonal α-Calnexin | Enzo Life Sciences | Cat#ADI-SPA-860-F RRID:AB_11178981 | WB(1:20,000) |
| Sequence-based reagent | Lox-gtF | Integrated DNA Technologies | PCR primers | CTG ACA CAG TGA GTG AAG GTG C |
| Sequenced-based reagent | Lox-gtR | Integrated DNA Technologies | PCR primers | GGT GCT GAG GAA GTG CTG TG |
| Sequenced-based reagent | Frt-gtF | Integrated DNA Technologies | PCR primers | AGG GGC CAT AGA GTG GTT AGG |
| Sequenced-based reagent | Frt-gtR | Integrated DNA Technologies | PCR primers | CTT AGC CGC TTT GTG CTG |
| Sequenced-based reagent | Rosa-gtF | Integrated DNA Technologies | PCR primers | AGT CGC TCT GAG TTG TTA TCA G |
| Sequenced-based reagent | Rosa-gtR | Integrated DNA Technologies | PCR primers | CTG ACA CAG TGA GTG AAG GTG C |
| Commercial assay or kit | ABC Kit | Vectastain | Cat#PK6100 RRID:AB_2336819 | Use kit directions |
| Commercial assay or kit | Prolong Gold Antifade | Thermo Scientific | Cat#P36930 | Use kit directions |
| Software, algorithm | Stereoinvestigator | MBF Bioscience | https://www.mbfbioscience.com/stereo-investigator _ | N/A |
| Software, algorithm | ImageJ | NIH | https://imagej.nih.gov/ij/; RRID:SCR_003070 | N/A |

*Continued on next page*

*Continued*

| Reagent type (species) or resource | Designation | Source or reference | Identifiers | Additional information |
|---|---|---|---|---|
| Software, algorithm | Prism | Graphpad | http://www.graphpad.com; RRID:SCR_002798 | N/A |
| Strain, strain background (*M. musculus*, male and female) | *Tor1b$^{+/-}$* | University of Connecticut Center for Mouse Genome Modification | N/A | B6;129 hybrid |
| Strain, strain background (*M. musculus*, male and female) | Tor1a$^{tm1Wtd}$/J | The Jackson Laboratory | RRID:IMSR_JAX:006251 | B6;129 hybrid |
| Strain, strain background (*M. musculus*, male and female) | Tor1a$^{tm3.1Wtd}$/J | The Jackson Laboratory | RRID:IMSR_JAX:025832 | B6;129 hybrid |
| Strain, strain background (*M. musculus*, male and female) | STOCK Tor1a$^{tm2Wtd}$/J | The Jackson Laboratory | RRID:IMSR_JAX:025637 | B6;129 hybrid |
| Strain, strain background (*M. musculus*, male and female) | *Tor1ab$^{\Delta E\ floxed\ Tor1b/+}$* | This paper (University of Connecticut Center for Mouse Genome Modification) | N/A | B6;129 hybrid Mouse line will be submitted and available from The Jackson Laboratory |
| Strain, strain background (*M. musculus*, male and female) | Emx1$^{tm1(cre)Krj}$/J | The Jackson Laboratory | RRID:IMSR_JAX:005628 | B6;129 hybrid |
| Strain, strain background (*M. musculus*, male and female) | Tg(dlx5a-cre)1Mekk/J | The Jackson Laboratory | RRID:IMSR_JAX:008199 | B6;129 hybrid |
| Strain, strain background (*M. musculus*, male and female) | Tg(Nes-cre)1kln/J | The Jackson Laboratory | RRID:IMSR_JAX:003771 | B6;129 hybrid |
| Strain, strain background (*M. musculus*, male and female) | B-OE | This paper (from Biocytogen) | N/A | B6;129 hybrid Mouse line will be submitted and available from The Jackson Laboratory |

## Generation and maintenance of mice

*Nestin-Cre*, *Emx1-Cre*, and *Dlx5/6-Cre* mice were obtained from the Jackson Laboratory. *Tor1b$^{+/-}$* and *Tor1a* ΔE floxed *Tor1b* mice were generated at the University of Connecticut Center for Mouse Genome Modification. Standard methods were used to engineer the *Tor1a* and *Tor1b* loci to include the ΔE mutation and loxP sites flanking exon 3–5 of *Tor1b*. The torsinB overexpression B-OE mouse line was generated with Biocytogen using CRISPR/Extreme Genome Editing technology, which increases homologous recombination efficiency over standard CRISPR/Cas9 technology. Mice were maintained in our mouse colonies at the University of Michigan and the University of Texas Southwestern Medical Center. Mice were genotyped for *Tor1a*, *Cre* recombinase, *Tor1b*, and *Tor1a/Tor1b* double floxed as previously described (*Liang et al., 2014*; *Tanabe et al., 2016*). Genotyping of novel lines (*Tor1a* ΔE floxed *Tor1b* and B-OE) is described in *Table 2* below. Primer sequences are located in the Key Resources Table.

The following breeding schemes were used to generate experimental animals and littermate *Cre* controls in our studies:

TorsinB KO: *Tor1b$^{+/-}$xTor1b$^{+/-}$*
A-CKO (Emx1): *Cre; Tor1a$^{+/-}$xTor1a$^{flx/flx}$*

**Table 2.** Genotyping PCR programs and band sizes.

| Gene | Primer name | Cycle | Band sizes |
|---|---|---|---|
| *Tor1a* ΔE floxed *Tor1b* | Lox-gtF | 94℃, 3 min; 94℃, 30 s; 61.5℃, 30 s; 72℃, 30 s, 34 cycles; 72℃, 5 min | WT – 245 bp<br>Floxed– 336 bp |
| | Lox-gtR | | |
| | Frt-gtF | | |
| | Frt-gtR | | |
| B-OE | Rosa-gtF | 94℃, 3 min; 94℃, 30 s; 61℃, 30 s; 72℃, 30 s, 30 cycles; 72℃, 5 min | WT – 469 bp<br>Mutant – 188 bp |
| | Rosa-gtR | | |
| | Mut-R | | |

A+B-CKO (Emx1): *Cre; Tor1ab$^{+/-}$xTor1a/Tor1b$^{flx/flx}$*
SKI;*Tor1b$^{+/+}$*(Emx1 and Nestin): *Cre; Tor1a$^{ΔE/+}$x Tor1a$^{flx/flx}$*
SKI;*Tor1b$^{+/-}$*(Emx1): *Cre; Tor1a$^{ΔE/+}$x Tor1a/Tor1b$^{flx/flx}$*
SKI;*Tor1b$^{-/-}$*(Emx1): *Cre; Tor1a/Tor1b$^{ΔE\ floxed\ torsinB/+}$x Tor1a/Tor1b$^{flx/flx}$*
B-OE (Dlx5/6 and Nestin): *Cre; Tor1a$^{+/-}$xTor1a$^{flx/flx}$; B-OE*.

Mice were housed in a temperature- and light-controlled room and provided access to food and water ad libitum. Mice of all genotypes were housed together to prevent environmental bias. Animals were housed 2–5/cage on a 12 hr light/dark cycle. All behavioral testing occurred during the light cycle. Age and sex-matched littermates were used as controls for all experiments. All controls unless noted otherwise were *Cre*+ controls. All procedures complied with national ethical guidelines regarding the use of rodents in scientific research and were approved by the University of Michigan and University of Texas Southwestern Institutional Animal Care and Use Committees.

## Immunohistochemistry

Immunohistochemistry in Emx1-*Cre* studies (*Figures 1–2*) was performed as previously described (*Liang et al., 2014*). Immunohistochemistry in torsinB rescue studies (*Figures 3– 5*) was performed as previously described (*Pappas et al., 2015*).

Mice were anesthetized with an intoxicating dose of ketamine xylazine and then transcardially perfused with 0.1 M phosphate buffered saline (PBS) followed by 4% paraformaldehyde in 0.1 M PB. Brains were extracted, postfixed in in 4% paraformaldehyde in 0.1 M overnight and then cryoprotected in 20% sucrose in 0.1 M PB. For *Emx1-Cre* studies (*Figures 1–2*), 16 µm sections were collected onto glass slides. Unstained tissue sections were permeablized using PBS containing 0.3% Triton X-100 followed by blocking in 10% normal donkey serum (NDS) for 1 hr at room temperature (RT). The sections were incubated with primary antibodies at 4℃ overnight (GFAP 1:2000, CUX1 1:200, CTIP2 1:500). Tissue sections were then washed and incubated with fluorophore conjugated secondary (1:1,000) antibodies for 2 hr at RT. Sections were mounted onto charged glass slides and coverslipped with prolong gold antifade mounting medium.

*Nestin-Cre* (*Figures 3,4*) and *Dlx5/6-Cre studies* (*Figure 5*), serial 40 µm sections were generated and stored in PBS. For fluorescence staining, free-floating sections were washed in PBS with 0.1% Triton X-100 (PBS-Tx), blocked in 5% NDS, and incubated overnight with primary antibody (GFAP 1:2000). Sections were then washed with PBS-Tx, and incubated with secondary antibodies (1:800) for 1 hr at RT. For DAB staining, sections were washed in PBS-Tx and endogenous peroxidase activity was extinguished by blocking with 0.3% H2O2. Sections were then blocked by 5% NDS and incubated in primary antibody (ChAT 1:200) overnight at 4℃. Sections were then washed with PBS-Tx, incubated in biotinylated secondary antibody (1:800) for 1 hr at RT, and then in avidin-biotin peroxidase complex (Vectastain Elite ABC Kit Standard; PK6100, Vector Laboratories, Burlingame, CA) for 1 hr at RT. Sections were then incubated with 3,3' diaminobenzidine (Sigma D4418) and quenched with PBS to stop further staining. Sections were then mounted onto charged glass slides, dried overnight, dehydrated in ascending ethanols, cleared in xylenes, and coverslipped with permount mounting medium.

For Nissl staining in all studies, brain sections mounted on gelatin-coated slides were dried overnight, rehydrated in descending ethanols, incubated for 3 min in 0.005% cresyl violet solution with

acetic acid, dehydrated in ascending ethanols, cleared in xylenes, and coverslipped with permount mounting medium.

## Cell counting and morphologic analysis

All cell counts and brain morphology measurements (striatal volume, cortical thickness) were performed by investigators blinded to genotype using a Zeiss Axio Imager M2 microscope. CUX1 and CTIP2 immunoreactive neurons were quantified in the sensorimotor cortex as follows. A 400 µm x 1,600 µm region of interest was generated using 16 µm sagittal sections for all animals at AP bregma + 0–2 mm and ML 1.20–1.32 mm. The images were viewed in ImageJ and cells expressing neuronal subtype markers were quantified by investigators blinded to genotype.

Stereology on striatal small and medium sized Nissl stained neurons and ChAT immunoreactive neurons was performed as previously described (*Pappas et al., 2015*). Cortical thickness and striatal volume were measured as previously described (*Pappas et al., 2015*). Brain area was measured in sagittal sections by creating a contour around the brain in the section corresponding to ML 1.44 mm (*Franklin and Paxinos, 2013*) and measuring area of the tracing using Stereoinvestigator.

## Western blotting

Mice were anesthetized using isoflurane and decapitated. Striatum, cerebellum, whole brain, or liver were dissected in tris buffered saline (TBS). Tissue was placed into microcentrifuge tubes with 150 µL lysis buffer (TBS with 1% sodium dodecyl sulfate, 1 mM Dithiothreitol, and Halt Protease Inhibitor Cocktail [Thermo Scientific #78430]) and homogenized using a plastic plunger. Supernatants were collected after centrifugation at 12,000 rpm for 10 min. Bradford protein assay was performed to assess protein concentration final lysates prepared in the same lysis buffer at 1 µg/µL including sample-loading buffer (Invitrogen NP0007). Samples were boiled for 5 min. 5 µg of protein and Dual Precision Plus protein standard (Bio-Rad #1610374) were loaded and run on 4–12% Bis-Tris gels (Invitrogen NP0323PK2). Protein was then transferred (350 mA for 1 hr at 4°C) onto a 0.22 µm PVDF membrane in transfer buffer with 10% methanol. Membranes were washed in TBS containing 0.1% Tween-20 (TBS-T). Membranes were blocked with 5% non-fat dry milk in TBS-T and incubated with primary antibody (torsinB 1:1,000, torsinA 1:10,000, calnexin 1:20,000) overnight at 4°C. Membranes were then washed with TBS-T and incubated with horseradish peroxidase-conjugated anti-rabbit secondary antibody (1:20,000) for 1 hr at RT. Bands were visualized using Supersignal West Pico PLUS chemiluminescent substrate (Thermo Scientific #34580) and exposed to CL-XPosure film (Thermo Scientific #34090). Developed films were scanned and protein bands were quantified using ImageJ.

## Tail suspension testing

Mice were picked up by the tail and suspended above the home cage for one minute while and video recorded by a camera. two observers blinded to experimental group graded videos for duration and presence of abnormal movements. Limb clasping was defined as a sustained abnormal posture with forepaws or hind paws tightly clasped together. Trunk twisting was defined as presence of twisting of the trunk to the point where both dorsal and ventral sides of the mouse are visible. Note that trunk twisting is distinct from lateral bending that control mice exhibit in an attempt to right themselves.

## Preweaning reflexes

Forelimb hang: Mice were suspended by the forelimbs using a 3 mm wire and the latency to fall was measured at P10 age. A cutoff of 30 s was used.

Negative geotaxis: Mice were placed on a wire grid, and the grid was tilted to an angle of 45° with the mice facing downward. Mice were assigned a score based on the following system: 3 – no movement, 2 – able to move but unable to turn around and face upward, 1 – able to turn and face upward but unable to climb the wire grid, and 0 – able to climb the wire grid. three trials were conducted with 30 s between trials on each designed day (ages P8, P10, P12).

Surface righting reflex: Mice were placed upside down on a flat surface and time to righting was measured using a stopwatch with a cutoff of 30 s. three trials were conducted on each testing day (P8, P10, P12) with 30 s of rest between trials.

## Statistics

All data sets are presented as mean ± SEM unless otherwise noted. Two-way ANOVAs, t-tests, and Chi square tests were performed using Graphpad prism software. Graphs were generated using the same software. Details regarding specific statistical tests, samples sizes, and p-values are located in the figure legends for each respective panel.

## Acknowledgements

We thank the members of the Dauer lab for their careful reading and suggestions for this manuscript. We also thank Christian Schlieker for the kind gift of the antibody to torsinB, Vivian Nguyen for technical assistance and Daniel Levin for careful behavioral analyses. This project was supported by: Tyler's Hope for a Dystonia Cure Foundation; NIH R01 NS077730; Bachmann Strauss Dystonia and Parkinson Foundation.

## Additional information

### Funding

| Funder | Grant reference number | Author |
| --- | --- | --- |
| Bachmann-Strauss Dystonia and Parkinson Foundation | | William T Dauer |
| National Institutes of Health | R01 NS077730 | William T Dauer |
| Tyler's Hope for a Dystonia Cure Foundation | | William T Dauer |

The funders had no role in study design, data collection and interpretation, or the decision to submit the work for publication.

### Author contributions

Jay Li, Conceptualization, Data curation, Formal analysis, Investigation, Methodology, Writing - original draft, Writing - review and editing; Chun-Chi Liang, Conceptualization, Data curation, Formal analysis, Investigation, Methodology, Writing - review and editing; Samuel S Pappas, Conceptualization, Supervision, Writing - original draft, Writing - review and editing; William T Dauer, Conceptualization, Resources, Supervision, Funding acquisition, Writing - review and editing

### Author ORCIDs

Jay Li https://orcid.org/0000-0002-8146-4450
Chun-Chi Liang http://orcid.org/0000-0002-8345-8564
Samuel S Pappas https://orcid.org/0000-0002-6980-2058
William T Dauer https://orcid.org/0000-0003-1775-7504

### Ethics

Animal experimentation: All procedures complied with national ethical guidelines regarding the use of rodents in scientific research and were approved by the University of Michigan (Protocol #00006600 and Protocol #00008870) and University of Texas Southwestern (Protocol #102767) Institutional Animal Care and Use Committees. Every effort was made to minimize both number of mice utilized as well as suffering.

### Decision letter and Author response

Decision letter https://doi.org/10.7554/eLife.54285.sa1
Author response https://doi.org/10.7554/eLife.54285.sa2

## Additional files

### Supplementary files
• Transparent reporting form

### Data availability
Our study did not generate sequencing or structural data. All source data files have been provided.

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
