## [Decision Letter]

**Acceptance summary:**

The data presented in this revised manuscript very nicely demonstrate that torsinB levels in the CNS modulate phenotypes derived from torsinA dysfunction in different mouse models. The implications, which are substantial, include that torsinB expression should be evaluated as a potential modifier of disease penetrance (only a third of DYT1 mutation carriers develop dystonia) and that strategies to enhance torsinB expression in neurons at the appropriate developmental time could provide significant therapeutic benefit. Overall, the described work is well done. The authors very nicely addressed concerns raised with the initial submission.

**Decision letter after peer review:**

Thank you for submitting your manuscript "TorsinB overexpression prevents abnormal twisting in DYT1 dystonia mouse models " for consideration by *eLife*. Your manuscript has been reviewed by three peer reviewers, one of whom is a member of our Board of Reviewing Editors, and the evaluation was overseen by Kate Wassum as the Senior Editor. The reviewers and the Reviewing Editor drafted this decision letter to help you prepare a revised submission.

Summary:

The work presented in this manuscript evaluates the role of torsinB expression as a potential modifier of DYT1 dystonia. DYT1 is caused by dominant mutations in TOR1A (encoding for the protein torsinA). The data very nicely demonstrate that torsinB levels in the CNS modulate phenotypes derived from torsinA dysfunction in different mouse models. The implications, which are substantial, include that torsinB expression should be evaluated as a potential modifier of disease penetrance (only a third of DYT1 mutation carriers develop dystonia) and that strategies to enhance torsinB expression in neurons at the appropriate developmental time could provide significant therapeutic benefit. Overall, the described work is well done. Yet, there are some areas needed to improve clarity prior to publication.

Essential revisions:

1) Characterization of the torsinB overexpression (Figure 3B) – there needs to be quantification of at least three samples with quantification shown in Figure 3 including the western blots. In addition, Figure 3 supplement n's/group need to be increased to 3 for statistical analysis.

2) Legend to Figure 3 needs to be corrected. Figure 3 has 8 images (A-H) yet the legend only describes A-G, skipping over the images of mice in Figure 3D.

3) TorsinA and B are very similar and seem to be expressed in the same cells. What is relative level of expression of torsinB to torsinA? The work would also be improved by showing further evidence of torsinB deletion by western blot or immunostaining.

4) The authors use multiple genetic mouse models in these studies to assess whether torsinB expression would impact behavioral and neuropathological phenotypes in models of DYT1. The description of the genetic makeup of the animal models used in the experiments throughout is somewhat confusing in the text and the figures. It is usually best that descriptions of the animals make clear the genetic modifications contained within the animal without having to decipher too much. When referring to the mice as Emx1-CKO and Emx1-dCKO, it is not clear what is conditionally knockout out. In general, the terminology is unclear for the different models and should be simplified, especially in the figures. In addition, Figure 1 is too small.

5) While it is appreciated that Emx1 expressing cortical pyramidal neurons project to the neurons in the striatum critically involved involved in basal ganglia function, is cortical dysfunction of these torsin genes implicated in dystonia? The assessment of the cortical phenotype in these models is performed well, it is unclear if the knockout of these genes in the cortex affects any striatal phenotypes? This is especially important given the focus in the field on striatal changes.

6) There is significant data provided on the clasping phenotype (such as worsening when torsinB is knocked out in the setting of torsinA mutants). It would be valuable to learn what other consequences on motor function this genetic interaction has. It is simply a worsening on clasping but otherwise motor function is similar? Or there is a significant worsening of motor function including abnormal spontaneous locomotion, rotarod, abnormal behavior in-cage observation, etc? With the current behavioral description, it is difficult for the reader to get a sense of how this interaction exacerbates phenotypes?

7) The behavior data should be included for the torsinB overexpression experiment in the *Nestin-Cre* background. This helps to demonstrate the functional change in these mice.

8) Is Dlx5/6 expressed in cholinergic neurons in the striatum? Is the observed change in striatal cholinergic neuron number cell autonomous?

9) In figure 3B, what is the exact genetic makeup of the animal shown in the Nes-Cre +, B-OE -, lane? Shouldn't there be a torsinB band in this lane?

10) Are the causes of cortical thinning with no changes on neuronal numbers due to reduced neuronal volume, reduced number of neurites, length of terminals, etc? This might help understand if/how torsinA function influences neuronal development of plasticity in a cell autonomous manner.

11) It would be helpful to know the neuronal versus glial contribution to torsinB expression in the overexpression model.

12) There is some evidence that overexpression torsinA might carry toxic consequences, making simply gene transfer as a therapeutic strategy challenging. Being a paralog, it is very likely that torsinB overexpression (i.e., AAV-driven) might not be ideal as it could also be toxic (in fact, the data presented here on mouse weight suggests this possibility). In the discussion, and based on what we currently know about torsinB, I challenge the authors to speculate how would they identify small molecules or design oligos that could de-repress or stabilize torsinB expression in CNS. Have they queried the Connectivity Map (CMap) for small molecular that upregulate torsinB? How do they propose to move forward? Similarly, how would they evaluate of torsinB expression moedulates penetrance or expressivity in patients and non-manifesting carriers? This would spice up a somewhat brief discussion.

13) Data concerning the behavior assessments that were described in the text as "data not shown" should be included.

---

## [Author Response]

Summary:The work presented in this manuscript evaluates the role of torsinB expression as a potential modifier of DYT1 dystonia. DYT1 is caused by dominant mutations in TOR1A (encoding for the protein torsinA). The data very nicely demonstrate that torsinB levels in the CNS modulate phenotypes derived from torsinA dysfunction in different mouse models. The implications, which are substantial, include that torsinB expression should be evaluated as a potential modifier of disease penetrance (only a third of DYT1 mutation carriers develop dystonia) and that strategies to enhance torsinB expression in neurons at the appropriate developmental time could provide significant therapeutic benefit. Overall, the described work is well done. Yet, there are some areas needed to improve clarity prior to publication.Essential revisions:1) Characterization of the torsinB overexpression (Figure 3B) – there needs to be quantification of at least three samples with quantification shown in Figure 3 including the western blots. In addition, Figure 3 supplement n's/group need to be increased to 3 for statistical analysis.

We agree that it is critical to further characterize torsinB overexpression. We have run new western blots with n = 3/genotype (Figure 3B). We simultaneously ran undiluted and diluted samples from torsinB overexpressing mice to enable quantification within the linear range. Assessment of these blots demonstrates approximately 30-fold increased torsinB expression via the torsinB overexpression allele as compared to controls (Figure 3C). We have also now repeated the western blots of torsinA in the brain and torsinB in the liver (Figure 3—figure supplement 1A-B and Figure 3—figure supplement 2A-B, respectively n = 3 / group).

2) Legend to Figure 3 needs to be corrected. Figure 3 has 8 images (A-H) yet the legend only describes A-G, skipping over the images of mice in Figure 3D.

We regret this oversight, and a correction has been made in the legend of Figure 3.

3) TorsinA and B are very similar and seem to be expressed in the same cells. What is relative level of expression of torsinB to torsinA?

TorsinA and torsinB appear to be expressed ubiquitously at different levels in different cell types. We previously reported that non-neuronal cells express significantly higher absolute levels of torsinB than torsinA, while absolute levels of torsinA are significantly higher in neurons (Kim et al., 2010.. Importantly, these quantitative immunoblotting experiments were performed only after normalizing conditions for anti-torsinA and anti-torsinB antibodies using lysates from cells expressing either GFP-torsinA or GFP-torsinB (allowing us to control for differing sensitivity of the antibodies).

The work would also be improved by showing further evidence of torsinB deletion by western blot or immunostaining.

We previously characterized the torsinB null and torsin (A+B) “double floxed” mouse lines by western blot analysis (Tanabe et al., 2016. The torsinB null mouse line achieves complete removal of torsinB protein, and the torsin (A+B) floxed line demonstrates selective deletion of torsinA and torsinB. We believe that these published data validate the torsinB null and torsin (A+B) floxed CKO models. We agree that immunohistochemical analysis of torsinB would be an excellent addition to enable more cell-type specific analyses. Unfortunately, in our experience, none of the currently available antibodies against torsinB are suitable for immunohistochemistry.

4) The authors use multiple genetic mouse models in these studies to assess whether torsinB expression would impact behavioral and neuropathological phenotypes in models of DYT1. The description of the genetic makeup of the animal models used in the experiments throughout is somewhat confusing in the text and the figures. It is usually best that descriptions of the animals make clear the genetic modifications contained within the animal without having to decipher too much. When referring to the mice as Emx1-CKO and Emx1-dCKO, it is not clear what is conditionally knockout out. In general, the terminology is unclear for the different models and should be simplified, especially in the figures. In addition, Figure 1 is too small.

We thank you for the feedback and agree that the nomenclature for multiple models made parts of the manuscript difficult to understand. We have reworked the model nomenclature to more directly highlight the manipulated genes. For example, our previous name of “Emx1-CKO” is now Emx1(A)-CKO, indicating that torsinA is deleted conditionally. The double conditional knockout mice are now Emx1(A+B)-CKO, indicating that both torsinA and torsinB are deleted conditionally. This system better highlights the torsin genes deleted in the Cre field of each model and improves clarity and readability.

We also adjusted the nomenclature in the gene dosage study (Figure 2). The mouse line that we previously called “Emx1-SKI-B2” (Emx1 selective knock-in mouse with 2 intact torsinB alleles), is now “Emx1-SKI;*Tor1b*^+/+^”, while mice lacking one or both torsinB alleles are now represented as “Emx1-SKI;*Tor1b*^+/-^” and “Emx1-SKI;*Tor1b*^-/-^,” respectively. We believe that these changes greatly improve the clarity of the different models. We also enlarged and reformatted Figure 1, as requested.

5) While it is appreciated that Emx1 expressing cortical pyramidal neurons project to the neurons in the striatum critically involved involved in basal ganglia function, is cortical dysfunction of these torsin genes implicated in dystonia? The assessment of the cortical phenotype in these models is performed well, it is unclear if the knockout of these genes in the cortex affects any striatal phenotypes? This is especially important given the focus in the field on striatal changes.

We have focused on several striatal phenotypes in the present study because they are robust, repeatable endpoints temporally linked with abnormal movements and with clear implications in disease. However, we did not intend to imply that dystonia is based solely in basal ganglia dysfunction. Indeed, clinical electrophysiological studies of dystonia patients (including DYT1) highlight several prominent cortical features such as deficient cortical inhibition, maladaptive cortical plasticity, and abnormal sensorimotor function (reviewed in Hallett et al., 2011), which drove our decision to target forebrain inhibitory neurons using the Dlx5/6-Cre line (Pappas et al., 2015; Pappas et al., 2014). The link between cortical function and dystonia symptoms has been clarified in the *Emx1-Cre* section.

We previously assessed gliosis, neurodegeneration, and several cell biological endpoints (*e.g.*, perinuclear ubiquitin accumulation and Hrd1 mislocalization) in the Emx1-Cre model (Liang et al., 2014. These deficits were highly selective to the cerebral cortex, with the most severe changes occurring in layer 5 of the sensorimotor cortex, and no changes observed in the striatum. These and all other data from our laboratory demonstrate that these phenotypes are restricted to the regions with torsinA knockout or mutant knock-in, strongly supporting a cell autonomous defect for these structural and molecular phenotypes. A discussion of cell-autonomous neurodegeneration has been included in the Discussion section. It is certainly feasible that disruption of cortical neurons could also result in abnormal striatal neuronal activity and dysfunction of basal ganglia output. We have not yet explored these possible functional changes in Emx1-Cre based models. Determination of the downstream functional effects of torsinA dysfunction is an important future direction, but one we believe is beyond the scope of the current study.

6) There is significant data provided on the clasping phenotype (such as worsening when torsinB is knocked out in the setting of torsinA mutants). It would be valuable to learn what other consequences on motor function this genetic interaction has. It is simply a worsening on clasping, but otherwise motor function is similar? Or there is a significant worsening of motor function including abnormal spontaneous locomotion, rotarod, abnormal behavior in-cage observation, etc? With the current behavioral description, it is difficult for the reader to get a sense of how this interaction exacerbates phenotypes?

Essentially all torsinA loss of function phenotypes appear to be exacerbated by removal of torsinB, including neurodegenerative, molecular, and organismal endpoints. Unfortunately, motor behavioral examination of these models is confounded by early lethality when both torsinA and torsinB are deleted. For example, we previously reported that deleting both torsinA and torsinB leads to lethality during development and maturation in several distinct models, including Dlx5/6-Cre, Synapsin-Cre, and *Nestin-Cre* lines (Tanabe et al., 2016; Figure S4, Table S1). Similarly, the Emx1-(A+B)-CKO model exhibits severe hypoactivity in the homecage, lack of growth or weight gain, delayed maturation, and appears moribund during the third week of life. Indeed, the University of Michigan IACUC determined that the Emx1-(A+B)-CKO model must be euthanized by P28 as a humane endpoint. While we are unable to perform open field or rotarod testing on these young and significantly smaller pups, we notice anecdotally that motor development appears to be delayed. We have added survival data and growth curves to this manuscript (Figure 1—figure supplement 3A-B).

7) The behavior data should be included for the torsinB overexpression experiment in the Nestin-Cre background. This helps to demonstrate the functional change in these mice.

This data has been included in table format in the revised manuscript (Figure 3—figure supplement 3A).

One point that we felt needed to be addressed in the manuscript is the ability of torsinB to rescue abnormal phenotypes in the presence of ΔE torsinA. The torsinB dosage study on the Emx1-SKI genetic background clearly demonstrates that torsinB reduction has a meaningful interaction in the presence of the disease allele, but it does not address rescue.

To date, no conclusive evidence has demonstrated that torsinA and torsinB function within the same complex. However, given the possibility that the torsinA ΔE allele exerts dominant negative effects, it is critical to address the ability of torsinB to rescue mutant torsinA phenotypes. We therefore added new data using the “Nes-SKI” model that expresses *Tor1a^ΔE/-^* in the nervous system (*Nestin-Cre*selective knockin; first characterized in Liang et al., 2014). TorsinB overexpression rescued Nes-SKI phenotypes including neurodegeneration, gliosis, and behavioral and developmental endpoints (Figure 4). These data demonstrate that torsinB can ameliorate mutant torsinA LOF phenotypes.

8) Is Dlx5/6 expressed in cholinergic neurons in the striatum? Is the observed change in striatal cholinergic neuron number cell autonomous?

Dlx5/6-Cre is expressed in progenitors that give rise to GABAergic and cholinergic neurons, including ChIs (Monory et al., 2006). We previously demonstrated a near complete loss of somatic torsinA immunoreactivity in the entire striatum of Dlx-CKO mice (Pappas et al., 2015). In ongoing work in the laboratory, we laser microdissected striatal ChIs from postnatal day 14 (P14) Dlx-CKO and control mice, finding a 5-fold reduction in torsinA mRNA (even with the “contamination” of afferent axon terminals of extra-striatal torsinA positive neurons that project onto ChI [i.e., centromedian/parafascicular thalamic neurons, corticostriatal neurons, and midbrain dopaminergic neurons]).

To address whether ChI neurodegeneration is cell autonomous, we previously reported characterization of a Chat-Cre *Tor1a*-CKO mouse line (Pappas et al., 2018 *eLife*). These cholinergic-selective knockout mice demonstrated the same temporal and spatial topographic pattern of cell loss in the dorsal striatum, while sparing ventral striatal and basal forebrain cholinergic neurons, despite similar extent and timing of torsinA loss in those populations (as demonstrated by immunostaining at P0). We have added a section to the Discussion section emphasizing the cell autonomous nature of torsinA-LOF mediated neurodegeneration.

9) In figure 3B, what is the exact genetic makeup of the animal shown in the Nes-Cre +, B-OE -, lane? Shouldn't there be a torsinB band in this lane?

We regret that the presented blot was unclear. The genotype of this control sample is *Nestin-Cre; Tor1a*^+/+^; *Tor1b*^+/+^. TorsinB is present and a band is visible at a higher exposure, however, due to gross overexpression of torsinB in the adjacent sample, we chose to display a shorter exposure.

To clarify this issue, we have updated Figure 3B to demonstrate torsinB expression in 3 samples from each genotype, including multiple dilutions of the torsinB overexpression lysates, and higher and lower exposures of the same membrane. This allowed us to better demonstrate and quantify the extent of torsinB overexpression without unintentionally masking the presence of torsinB in control lysates.

10) Are the causes of cortical thinning with no changes on neuronal numbers due to reduced neuronal volume, reduced number of neurites, length of terminals, etc? This might help understand if/how torsinA function influences neuronal development of plasticity in a cell autonomous manner.

We have conducted an assessment of neuronal volume in Emx1-CKO mice. We observe no changes to average soma size. These data have been added as Figure 1—figure supplement 4A and suggest that gross changes to neuronal soma morphology may not contribute to cortical thinning.

We previously observed reactive gliosis in layer V of Emx1-CKO and Emx1-SKI mice (Liang et al., 2014). While there is not a statistically significant reduction in overall CTIP2+ neuron count in the Emx1-CKO model, we do observe a statistically significant reduction of ~10% in CTIP2+ neurons in layer Vb. This data has been incorporated into the manuscript as Figure 1—figure supplement 4B. This highly selective neurodegeneration may contribute to the cortical thinning observed in Emx1-CKO mice.

11) It would be helpful to know the neuronal versus glial contribution to torsinB expression in the overexpression model.

We employed a floxed-stop Rosa26-locus torsinB strategy, enabling us to overexpress in a variety of cell types based on the *Cre* used to activate the overexpression allele. *Nestin-Cre* is expressed in neural progenitor cells that give rise to both neurons and glia (Tronche et al., 1999). In contrast, *Dlx5/6-Cre* is active in the cells that give rise to forebrain GABAergic and cholinergic neurons, and is exclusive to neurons (Monory et al., 2006). The complete rescue observed with neuronal selective torsinB overexpression demonstrates that glial overexpression is dispensable for rescue in our models. In the revised manuscript, we address neuron vs. glia and cell autonomous degeneration/rescue in the Discussion section.

12) There is some evidence that overexpression torsinA might carry toxic consequences, making simply gene transfer as a therapeutic strategy challenging. Being a paralog, it is very likely that torsinB overexpression (i.e., AAV-driven) might not be ideal as it could also be toxic (in fact, the data presented here on mouse weight suggests this possibility). In the discussion, and based on what we currently know about torsinB, I challenge the authors to speculate how would they identify small molecules or design oligos that could de-repress or stabilize torsinB expression in CNS. Have they queried the Connectivity Map (CMap) for small molecular that upregulate torsinB? How do they propose to move forward? Similarly, how would they evaluate of torsinB expression moedulates penetrance or expressivity in patients and non-manifesting carriers? This would spice up a somewhat brief discussion.

We agree that these are critical points. Two clear future directions of this work are to identify safe translational strategies for torsinB supplementation and to study the role of torsinB in dystonia penetrance and expressivity.

We are currently developing a torsinB-expressing AAV strategy, with the goal of titrating the expression of torsinB levels to determine a therapeutic window for rescue of torsinA LOF. Based on extensive experience with these models, we anticipate that a much lower level of overexpression will be sufficient for rescue. We are also in the process of developing compounds and identifying pathways that modulate torsinB expression, primarily via screening technologies in cells modified to report torsinB expression. Nevertheless, we believe that a genetic approach – e.g. an mRNA stabilizing ASO, or CRISPR-based torsinB activation – is likely to the approach that is most efficacious and least likely to have off target effects.

The potential role of torsinB in dystonia penetrance and expressivity is an interesting concept that we have grappled with but is technically challenging. Postmortem samples from patients with inherited dystonia are exceedingly rare and poorly annotated, essentially eliminating the possibility of assessing the relative levels of torsinA and torsinB in neural tissue. Moreover, our model suggests that the levels during childhood are the key determinants, raising questions about the relevance of protein levels decades later. To potentially overcome these issues, we are collecting fibroblast samples from manifesting, non-manifesting and control subjects, and will assess mRNA and proteins levels during neural maturation. Our revised manuscript now addresses these points in the Discussion section.

13) Data concerning the behavior assessments that were described in the text as "data not shown" should be included.

We added tables of limb clasping in torsinB null mice (Figure 1—figure supplement 1E) and of *Nestin-Cre*-(A)-CKO phenotypes (Figure 3—figure supplement 3A).